# Notch1 promotes ordered revascularization through Semaphorin 3g modulation of downstream vascular patterning signalling factors

James Hyun[1], Monica Lee[2], Jalees Rehman[1], Kostandin V. Pajcini[1] and Asrar B. Malik[1]

[1]*Department of Pharmacology and Regenerative Medicine, University of Illinois College of Medicine, Chicago, IL, USA*
[2]*Department of Physiology and Biophysics, University of Illinois College of Medicine, Chicago, IL, USA*

Edited by: Don Bers & Livia Hool

The peer review history is available in the Supporting Information section of this article (https://doi.org/10.1113/10.1113/JP282286#support-information-section).

**Abstract** Here we genetically and functionally addressed potential pathways of Notch signalling in mediating vascular regeneration in mouse models. We first used transgenic adult mice with either gain- or loss-of-function Notch signalling in vascular endothelial cells and monitored perfusion in the hindlimb following ischaemia induced by femoral artery ligation. Mice deficient in Notch

**James Hyun** received his PhD from UIC which focused on endothelial cell responses under ischaemia. He is currently a postdoctoral fellow at UCLA furthering his training in vascular biology.

signalling showed defective perfusion recovery and expansion of collateral arteries. Transcriptomics analysis of arterial endothelial cells in the Notch mutants identified the guidance factor *Sema3g* as a candidate gene mediating reperfusion downstream of Notch. Studies in the retinal circulation showed the central role of SEMA3G downstream of Notch signalling in the orderly regulation of vascular patterning. These studies in multiple vascular beds show the primacy of Notch signalling and downstream generation of guidance peptides such as SEMA3G in promoting well-ordered vascular regeneration.

(Received 27 August 2021; accepted after revision 29 November 2021; first published online 17 December 2021)

**Corresponding authors** Dr Kostandin V. Pajcini and Dr Asrar B. Malik: Department of Pharmacology and Regenerative Medicine, University of Illinois College of Medicine, Chicago, IL 60607, USA.    Emails: kvp@uic.edu, abmalik@uic.edu

**Abstract figure legend** A feedback mechanism during vascular regeneration permits organization and expansion of collateral arteries is initiated by canonical Notch1 signaling which activates the expression of Sema3g that in turn inhibits Vegf-mediated angiogenesis.

## Key points

- Notch signalling is a critical mediator of revascularization. Yet the cellular processes activated during recovery following vascular injury are incompletely understood. Here we used genetic and cellular approaches in two different vascular beds and cultured endothelial cells to address the generalizability of mechanisms.
- By utilizing a highly reproducible murine model of hindlimb ischaemia in transgenic mice in which Notch signalling was inhibited at the transcriptional level, we demonstrated the centrality of Notch signalling in perfusion recovery and revascularization.
- RNA-sequencing of Notch mutants identified class 3 Semaphorins regulated by Notch signalling as downstream targets. Studies in retinal vessels and endothelial cells showed an essential role of guidance peptide Sema3g as a modulator of angiogenesis and orderly vascular patterning.
- The Notch to Sema3g signalling axis functions as a feedback mechanism to sculpt the growing vasculature in multiple beds.

## Introduction

Revascularization has focused on enhancing vascular endothelial growth factor (VEGF) signalling and activation of downstream pathways (Takeshita *et al.* 1994; Losordo *et al.* 1999; Webber *et al.* 2011). VEGF is highly upregulated in tissue following critical events such as vaso-occlusion, myocardial infarction-associated ischaemia and stroke (Banai *et al.* 1994; Li *et al.* 1996), and during developmental vascular network formation in the retina (Stalmans *et al.* 2002). VEGF binding to its receptor VEGFR2 drives the angiogenic programme leading to tissue reperfusion and oxygenation (Lampugnani *et al.* 2006; Kawamura *et al.* 2008; Chen *et al.* 2010). VEGF generated in endothelial cells activates Notch signalling, which is required for the regulation of vascularization and vascular patterning (Roca & Adams, 2007; Simons & Eichmann, 2015). In the current understanding, a VEGF gradient establishes endothelial cells to compete for tip or stalk positions to grow outwards to form fully functional new vessels (Blanco & Gerhardt, 2013a). The concentration of VEGF dictates which endothelial cells adopt the tip cell position and which ones upregulate the Notch ligand Delta-like 4 (Dll4). Cells expressing Dll4 signal activate the Notch1 receptor in neighbouring cells and induce proliferation, migration and formation of a vascular network (Siekmann & Lawson, 2007; Blanco & Gerhardt, 2013b; Travisano *et al.* 2019). Additionally, Notch signalling functions in conjunction with the Ets, Sox and Fox family of developmental transcription factors to establish specific components of the vasculature, the arterial, venous and lymphatic systems (Seo *et al.* 2006; Kataoka *et al.* 2011; Corada *et al.* 2013). Recent studies have identified the critical role of neuronal guidance molecules such as Semaphorins in directing vascular formation (Liu *et al.* 2016; Chen *et al.* 2021).

Although Notch signalling is central to endothelial development and developmental vascular generation, its involvement in adult arterial vessel regeneration is less well understood. Endothelial-specific deletion of the Notch1 ligand Dll4 postnatally showed reduced perfusion due to defective angiogenesis following hind-limb ischaemia (Takeshita *et al.* 2007; Cristofaro *et al.*

2013). Loss of other Notch ligands such as Dll1 also exhibited reduced perfusion due to poor macrophage differentiation which inhibited expansion of collateral arteries (Limbourg *et al.* 2007; Krishnasamy *et al.* 2017). In addition, loss of Notch signalling decreased nitric oxide production in endothelial cells, also contributing to reduced revascularization (Chang *et al.* 2013). These findings collectively show an especially important role for Notch signalling in endothelial regeneration and revascularization. However, the downstream pathways mediating orderly vascularization and patterning, their relationship to Notch, and the generalizability of the signals to multiple vascular beds remain unclear.

Notch receptors are classified as single-pass surface proteins that initiate a panoply of transcriptional programmes and activate downstream gene targets mediating reperfusion. Utilizing a hind-limb ischaemia model in combination with inducible *in vivo* transgenic murine models for inhibiting Notch signalling using Dominant Negative Mastermind (DNMAML) and hyperactivation of Notch through overexpression of Notch intracellular domain (NICD), we identified Semaphorin3g (Sema3g) as a key downstream target of Notch during vascular regeneration. We also showed Sema3g functioned as a modulator of vascularization and contributed to the patterning of the vascular network.

## Materials and methods

### Ethical approval

Animals were bred and maintained in a pathogen-free setting at the University of Illinois at Chicago following approval by the Institutional Animal Care and Use Committee. Animal experiments were conducted in accordance with NIH guidelines for the care and use of laboratory animals. Animal experiments were conducted under anaesthesia induced by ketamine (75 mg/kg) and xylazine (15 mg/kg). Subcutaneous injection of bupivacaine was given as pre-operative analgesia near the incision site. Bupernorphine SR (1 mg/kg) was given as post-operative analgesia. Mice were monitored daily for signs of illness, reluctance to move, eat or drink, hunched posture and weight loss. Animals showing signs of distress were killed and excluded from the study. Animals were killed by $CO_2$, where a chamber was filled with 100% $CO_2$ at a fill rate of 4 l/min and maintained for an additional minute once the animal had become unconscious and respiratory arrest conducive of death by $CO_2$ exposure. A secondary method of cervical dislocation was then used.

### Mouse strains and tamoxifen injections

All transgenic mice used in these studies were of C57BL/6J background aged 8–12 weeks and had access to food and water *ad libitum*. Postnatal inhibition of Notch signalling and overexpression of NICD was restricted to the endothelium by the generation of DNMAML^f/f:*Cdh5-CreERT2* and ICN^f/f:*Cdh5-CreERT2* mice. DNMAML^f/f and ICN^f/f were provided by Dr Warren S. Pear (University of Pennsylvania). *Cdh5-CreERT2* mice were provided by Dr Ralf Adams (Max Planck Institute). Reporter mice used in RNA sequencing (RNAseq) studies (ROSAmT/mG mice; Jackson Lab, Bar Harbor, ME, USA; Stock #007576) were also crossed with *Cdh5-CreERT2* mice. To induce transgene expression mice were injected intraperitoneally at 8 weeks of age with tamoxifen, 75 mg/kg dissolved in corn oil, for 4 days and left to recover for 1 week before experimental procedures. Cre-negative littermates were used as controls and were injected with tamoxifen under the same regiment. One gram of tamoxifen (Sigma, St Louis, MO, USA; cat# T5648-1G) was dissolved into 50 ml of corn oil (Sigma, cat# C8267) by shaking overnight at room temperature.

### Ischaemia model and laser Doppler blood flow measurements

Animal experiments were carried out according to approved protocols by the Animal Care Committee (ACC) at the University of Illinois, Chicago. Twenty mice from each genotype, DNMAML, ICN and control littermates, were age and sex matched before being subjected to hindlimb ischaemia. Anaesthesia was induced by I.P. administration of ketamine (75 mg/kg) and xylazine (15 mg/kg). Unilateral ischaemia was performed by ligation and resection of the left femoral artery, proximal to the epigastric artery and distal to the proximal caudal femoral artery, using 6-0 silk sutures; neighbouring arterial branches were cauterized using an electrical coagulator. Wounds were closed using 5-0 nylon sutures and care was taken not to disturb the femoral nerve. Mice where the femoral vein was disturbed and ruptured were killed and not included in the study. Bupernorphine SR (ZooPharm, Wymoing, USA; cat#1Z-7300) was given (1 mg/kg) for post-operative analgesia. Depth of anaesthesia was assessed by muscular relaxation and toe pinch every 10 min during the procedure. Mice were given additional ketamine (37 mg/kg) and xylazine (7.5 mg/kg) to sustain anaesthesia or to increase the depth of anaesthesia. Mice groups were randomized and perfusion measurements of hindlimbs were obtained using a laser Doppler blood flow (LDBF) analyser (Perimed AB, Järfälla, Sweden; PeriScan PIM 3 system). Mice were anaesthetized using 1% isoflurane (Piramal Enterprises Ltd.) and equilibrated on a heating pad for 10 min. Hindlimbs were identified and outlined before blood flow measurements. Analysis of blood flow was assessed by capturing the mean

blood flow of ischaemic (surgical) and non-ischaemic (contralateral) limbs and expressed as ratios of ischaemic to non-ischemic LDBF. LDBF was displayed as changes in laser frequencies by different colour pixels and mean LDBF values were used for analysis.

### Immunohistochemistry and histological assessment

Adductor muscles of ischaemic and non-ischaemic hind-limbs were excised on day 7 following hindlimb ischaemia. Mice were given a lethal dose of ketamine/xylazine (150 mg/kg) before being perfused with 10 ml PBS through the descending aorta at a flow rate of 10–15 mL per min, and 10 ml of 10% neutral buffered formalin (NBF) was subsequently perfused to fix the vasculature. Adductor muscles were then isolated and post-fixed for an additional 14 h in 10% NBF and transferred to 70% ethanol before being embedded in paraffin as described in Nature protocol 2009 (Limbourg et al. 2009). Serial tissues slices were cut at 5 $\mu$m $\mu$ thickness and samples were hydrated, and antigen was retrieved with Trilogy buffer (Sigma, cat# 920P-05) before immunohistochemistry experiments. Tissue sections were blocked with 10% donkey serum (Sigma, cat# D9663) for 1 h and primary antibodies were incubated overnight at 4°C. Slides were washed three times for 10 min in PBS and incubated with corresponding secondary fluorescent antibodies for 1 h. Tissues were washed with an additional 3 × 10 min PBS washes before being mounted with a cover slip containing ProLong Gold antifade mountant with DAPI (Invitrogen, Carlsbad, CA, USA; cat# P36935). Adductor muscle tissue sections containing semimembranosus muscles were imaged using an Olympus BX51 microscope at 40× magnification. The neurovascular bundle containing CD31+/a-SMA+ collateral arteries within the semi-membranosus muscle was identified and measured based on Limbourg *et al.* (2009) Positive fluorescence signal was determined using sections containing secondary antibodies only.

### Flow cytometry and cell sorting

Mice were killed under isoflurane and adductor muscles of ischaemic and contralateral limbs were excised. Tissues were mechanically digested using scissors and then enzymatically digested with Type1 collagenase I (1 mg/ml) in HBSS for 30 min at 37°C in a water bath. Cells were washed and centrifuged to remove collagenase for down-stream applications. Cells were treated with ACK lysis buffer (Gibco A10492-01) for 5 min at room temperature to lyse blood cells before being passed through a 70 and 40 $\mu$m filter to obtain a single cell suspension of cells. Cells were incubated with FC block on ice for 10 min before being stained with the fluorophore-conjugated antibodies listed below for 1 h on ice. Cells were washed with FACS buffer (PBS+1% FBS) containing DAPI to stain dead cells. Arterial endothelial cells were sorted using Moflow astrios and identified based on the cell surface staining of Cd45−/Ter119−/Cd31+/Sca1+/Pdpn− and analysed using Flowjo.

### Bulk RNA sequencing

Total RNA was extracted following the Trizol Reagent User Guide (Thermo Fisher Scientific, Waltham, MA, USA). One microlitre of 10 mg/ml glycogen was added to the supernatant to increase RNA recovery. RNA was quantified using a Qubit 2.0 Fluorometer (Life Technologies, Carlsbad, CA, USA) and RNA integrity was checked with TapeStation (Agilent Technologies, Palo Alto, CA, USA) to see if the concentration met the requirements.

### RNA library preparation and multiplexing

RNA samples were quantified using a Qubit 2.0 Fluoro-meter (Life Technologies) and RNA integrity was checked with a 2100 TapeStation (Agilent Technologies). RNA library preparations, sequencing reactions and initial bioinformatics analysis were conducted at GENEWIZ, LLC. (South Plainfield, NJ, USA). SMART-Seq v4 Ultra Low Input Kit for Sequencing was used for full-length cDNA synthesis and amplification (Clontech, Mountain View, CA, USA), and Illumina Nextera XT library was used for sequencing library preparation. Briefly, cDNA was fragmented and adaptor was added using transposase, followed by limited-cycle PCR to enrich and add index to the cDNA fragments. The final library was assessed with a Qubit 2.0 Fluorometer and Agilent TapeStation.

### Sequencing 2 × 150 bp PE

The sequencing libraries were multiplexed and clustered on two lanes of a flowcell. After clustering, the flowcells were loaded on the Illumina HiSeq instrument according to the manufacturer's instructions. The samples were sequenced using a 2 × 150 bp paired end (PE) configuration. Image analysis and base calling were conducted by the HiSeq Control Software (HCS) on the HiSeq instrument. Raw sequence data (.bcl files) generated from the Illumina HiSeq were converted to fastq files and de-multiplexed using the Illumina bcl2fastq v.2.17 program. One mismatch was allowed for index sequence identification.

### Data analysis

After demultiplexing, sequence data were checked for overall quality and yield. Then, sequence reads were

trimmed to remove possible adapter sequences and nucleotides with poor quality using Trimmomatic v.0.36. The trimmed reads were mapped to the *Mus musculus* mm10 reference genome available on ENSEMBL using the STAR aligner v.2.5.2b. The STAR aligner uses a splice aligner that detects splice junctions and incorporates them to help align the entire read sequences. BAM files were generated as a result of this step. Unique gene hit counts were calculated by using featureCounts from the Subread package v.1.5.2. Only unique reads that fell within exon regions were counted. After extraction of gene hit counts, the gene hit counts table was used for downstream differential expression analysis. Using DESeq2, a comparison of gene expression between the groups of samples was performed. The Wald test was used to generate *P*-values and log$_2$ fold changes. Genes with adjusted *P*-values <0.05 and absolute log$_2$ fold changes >1 were called as differentially expressed genes for each comparison. A principal components analysis (PCA) was performed using the 'plotPCA' function within the DESeq2 R package. The plot shows the samples in a 2D plane spanned by their first two principal components. The top 500 genes, selected by the highest row variance, were used to generate the plot. Additionally, the differentially expressed genes from the comparisons WT-B-vs-WT-A, DN-B-vs-DN-A and ICN-B-vs-ICN-A were extracted. Then, using the merge function in R, all the common differentially expressed genes between these three comparisons were found and a file was generated containing these genes.

### RNA isolation and qRT-PCR analysis

For qRT-PCR experiments, cells were harvested using TRIzol (Invitrogen, #15596026) at the manufacturer's recommended volumes. RNA was then isolated using a Direct-zol RNA microprep kit (Zymo Research, Irvine, CA, USA) as per the manufacturer's instructions. One microgram of cDNA was synthesized using random hexamers (Invitrogen, cat# N8080127) and Super-ScriptIII reverse transcriptase (Invitrogen, cat# 18080093) according to the manufacturer's protocol. Real time PCR experiments was performed on ABI Prism 7900HT systems with 2x SYBR Green PCR Master Mix (Applied Biosystems, cat# 4309155). All Ct values were normalized to *Ef1a* or *18S*, and primer information is listed in the Supplementary Methods.

### GSI (γ-secretase inhibitor) washout experiment

Low-passage human pulmonary arterial endothelial cells (HPAECs) were obtained from Lonza from a single donor (#C2517A) and cultured in six-well plates with EGM-2 (Cat CC-3156 and CC-4147). Cells were treated with 1 $\mu$M DAPT (Sigma, 208255-80-05) for 48 h before an 8 h washout experiment. Cells were harvested at the end of the experiment using TRIzol (Invitrogen, #15596026) for RNA isolation and qRT-PCR experiments.

### Lentiviral production

293FT cells (Invitrogen, cat# R70007), less than 50 passages, were plated at 50% confluence in a six-well tissue culture plate and cultured in DMEM (Gibco, cat# 11995065) supplemented with 10% serum overnight in normoxia at 37°C, 5% $CO_2$ and 21% $O_2$. Cells were washed with PBS and transfected using Lipofectamine 2000 (Invitrogen) as per the manufacturer's protocol containing lentiviral packaging plasmid psPAX2 (Addgene, Watertwon, MA, USA; plasmid#12260), pMD2.G (Addgene, plasmid#12259) and lentiviral overexpression plasmids pCCL-DNMAML and pCCL-GFP. Viral supernatant was collected daily and pooled. Lenti-X Concentrator (Takara, Shiga, Japan; cat# 631231) was used to concentrate lentiviral particles ten-fold, aliquoted and stored at −80°C. Lentivirus was titrated on HPAECs and a dose based on green fluorescent protein (GFP) expression flow cytometry, and a titre that resulted in <80% of cells being GFP-positive was used for experiments.

### SEMA3G expression and conditioned media

SEMA3G cDNA was purchased from Horizon Discovery MGC cDNA (cat# MHS6278-211689602) and PCR-amplified containing a C-terminal 3xflag sequence. The transgene was subsequently subcloned into pCDNA3 and the sequence verified before being transiently transfected (Invitrogen Lipofectamine 2000) into HEK 293T cells as per the manufacturer's protocol for 10 cm tissue culture dishes. Twenty-four hours after transfection, cells were cultured on Opti-MEM for 5 days in the presence of 100 $\mu$M Furin Inhibitor II (Sigma, SCP0148-5MG) and media were collected daily and stored at 4°C. Media were then pooled, buffer exchanged into 1× PBS pH 7.4 (Gibco, cat# 10010023), and concentrated 10× fold using a 50 kDa filter concentrator (Vivaspin500). SEMA3G containing conditioned media was validated by western blot analysis using anti-Flag antibody (Sigma, cat# F1804) as well as anti-SEMA3G antibody (Invitrogen, cat# PA5-51631) at the manufacturer's recommended dilutions.

### Endothelial 3D spheroid assay

Endothelial cell spheroids to test for sprouting angiogenesis were used as described by Heiss *et al.* (2015) with minor modifications. HPAECs expressing DNMAML or GFP were cultured at 37°C in a 5% $CO_2$, 21% $O_2$ incubator. Cells were detached using TrypLE

(Gibco, cat#12604-013) as per the manufacturer's protocol and made into a single cell suspension of EGM-2 (Lonza, cat# CC-3162). In total, 100,000 cells were transferred in a 15 ml conical flask containing 4 ml EGM-2 and 1 ml methyl cellulose solution. The cell solution was carefully mixed to avoid foaming and transferred to a sterile reagent reservoir. A multiple-channel pipette was used to transfer 25 $\mu$l of the cell mixture containing ~500 cells onto a 10 cm non-tissue culture plate. Cells were cultured upside down overnight at 37°C in a 5% $CO_2$, 21% $O_2$ incubator to form spheroids. One hundred spheroids were gently harvested using sterile PBS centrifuged for 5 min at 500 $g$ without breaking to pellet cells. Spheroids were gently resuspended with 2 ml of stock Collagen1 solution (Corning, cat#CB40236) supplemented with 10× DMEM (Sigma, cat#D2429), 10% FBS and 10% methylcellulose solution. Spheroids were equally divided (~500 $\mu$l) and transferred to a pre-warmed 24-well plate and left to undisturbed for 30 min to solidify in a 37°C, 5% $CO_2$, 21% $O_2$ incubator. Collagen gels containing endothelial spheroids were overlaid with 400 $\mu$l of EGM2 supplemented with 100 $\mu$l of condition medium containing SEMA3G or PBS vehicle. Spheroids were incubated overnight, and individual spheroids were imaged using a microscope and fluorescence magnification. Sprout lengths and area were analysed using ImageJ software and values were expressed in pixels and arbitrary units. Sprouting area was calculated by first setting the threshold to highlight areas that are GFP-positive. GFP-positive areas were measured and represented as a percentage based on the image size (2048 × 1536 pixels)

### Western blotting

DNMAML-expressing HPAECs and control HPAECs (100,000 in each case) were plated overnight in six-well plates and serum starved with EBM (Lonza, cat#CC-3121). Then, 50 ng/$\mu$l of recombinant VEGF$_{165}$ (PeproTech #100-20) was added to the medium for 3 and 10 min before the cells were washed with PBS and collected in RIPA buffer for western blotting. VEGFR2 competition studies were conducted identically, except for pre-treatment of 50 ng/$\mu$l recombinant SEMA3G for 10 min before the addition of VEGF$_{165}$. Expression of DNMAML was confirmed by immunoblotting of flag protein. Immunoblots were analysed for optical density using ImageJ software.

### Retinal angiogenesis assay

Angiogenesis of whole mount retinas was analysed based on previous studies with minor modifications (Pitulescu *et al.* 2010). Tissue-specific expression of DNMAML was induced through daily intragastric injections of tamoxifen starting at P1 and carried out until P3. Recombinant SEMA3G (Novus Bio, cat# H00056920) was purchased and resuspended in sterile PBS to a concentration of 10 $\mu$g/$\mu$l and injected daily intradermally near the right jugular vein starting at P4 until P5. DNMAML- and Cre-negative littermates were killed at P6 and eyes were harvested for staining experiments. Eyes were fixed in 4% PFA on ice for 20 min and stored in PBS before dissection. Retinas were isolated and hyaloid vessels were carefully removed under a dissection microscope. Retinas were then blocked/permeabilized for several hours in Pblec solution (1% Triton X-100, 1 mM $MgCl_2$, 1 mM $CaCl_2$, 1 mM $MnCl_2$, 1% BSA). The vasculature was stained by incubating retinas with Isolectin from *Griffonia simplicifolia* conjugated to Alexa Fluor 568 (Invitrogen, #I21412) for 3 h at 1:200 dilution in Pblec solution. Retinas were then washed with 4 × 15 min washes in Pblec and then 4 × 15 min washes in PBS before being mounted on a microscope slide with Vectashield (Vector labs, cat# H-1000-10). Slides were stored at 4°C before being imaged using an Olympus BX51 fluorescence microscope. High-resolution images (10× magnification) were imaged and tiled together using Adobe Illustrator and each leaflet containing angiogenic fronts was analysed. The number of endothelial extensions and vascular coverage was quantified using ImageJ. To determine vascular coverage, angiogenic fronts were identified and a threshold was set to visualize just the retinal vasculature. Coverage of isolectin-stained vasculature was then measured in pixels, in arbitrary units, and represented as a percentage based on the size of angiogenic front highlighted.

### Statistics

All results are expressed as mean $\pm$ SEM and were analysed using GraphPad Prism. The number of biological replicates and statistical tests used for analysis are given in the figure legends. Student's *t* test was used to analyse two experimental groups and one-way ANOVA was used to determine statistical differences involving three experimental groups, as specified in the figure legends. *P* values of <0.05 were considered statistically significant.

### Results

### Notch signalling is an essential feature of adaptive vascularization

Although the transcription factors *Hes1* and *Hey1* are downstream targets of Notch, the abundance of canonical RBPJ-binding sites on DNA suggests that relatively few functional Notch-dependent targets have thus far been identified (Wang *et al.* 2014). Here we identified potential

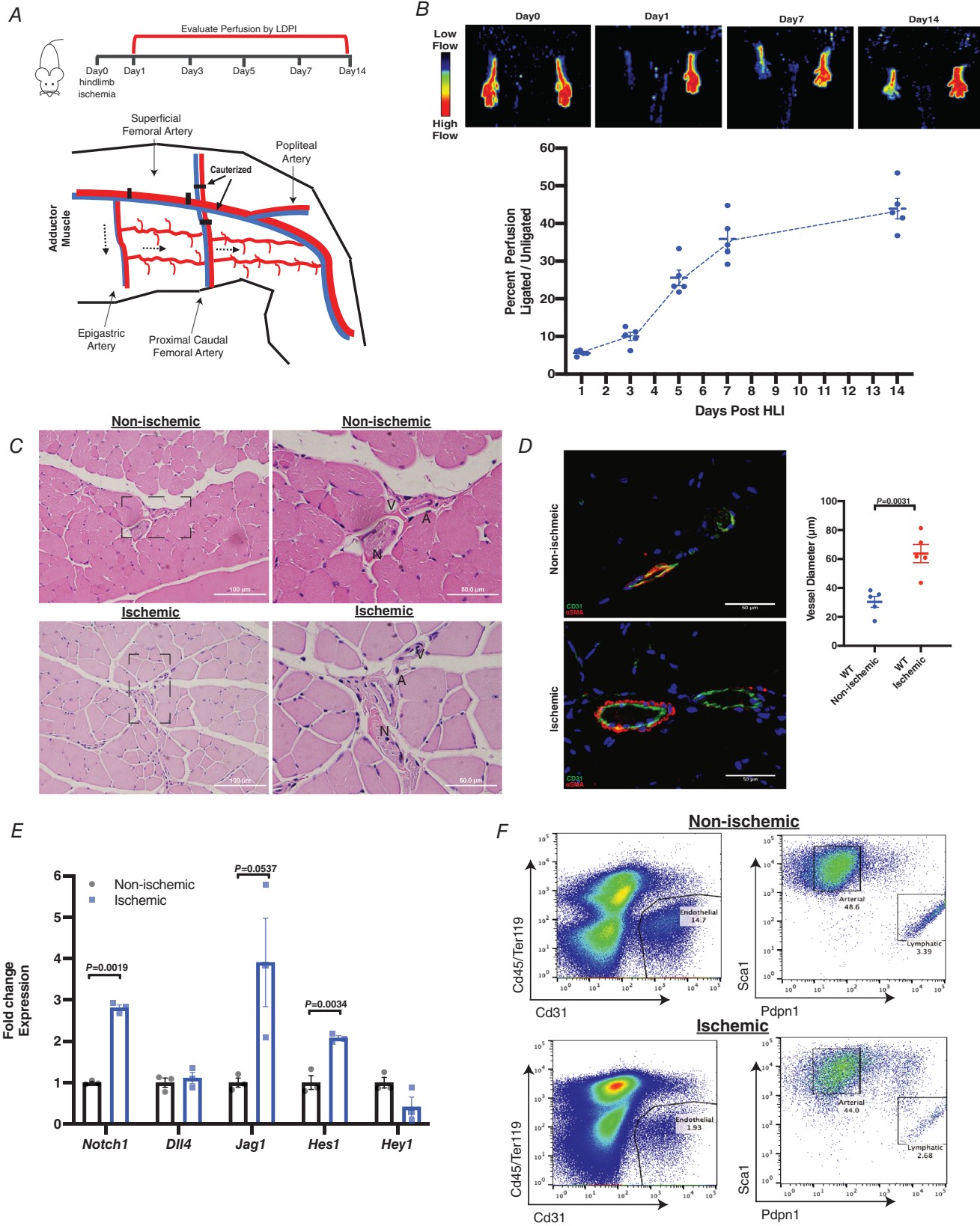

**Figure 1. Upregulation of endothelial Notch signalling following hindlimb ischaemia in mice**

*A*, schematic overview of hindlimb ischaemia (HLI) experiments. Mice (8–10 weeks old) underwent unilateral HLI. Perfusion of hindlimbs was monitored by laser Doppler imaging over 14 days following surgery. The superficial

femoral artery was carefully exposed and ligated at two locations before being excised. Neighbouring arterial beds were cauterized (black arrow) to re-route blood flow (dashed arrows) to collateral arteries in the adductor muscle. *B*, representative laser Doppler images of C57BL6J (WT) mice before (Day 0) and at indicated time points following hindlimb ischaemia. Images represent perfusion of blood into hindlimbs. Superficial blood in hindlimbs causes Doppler shifts that can be quantified and displayed as red to blue colours, where red indicates more blood flow than blue. Perfusion of hindlimbs was calculated using laser Doppler perfusion images (LDPIs) of ischaemic limbs and contralateral non-ischaemic limbs. Mean LDPIs of ligated limbs were divided by contralateral limbs and are represented as a percentage for each day over the course of 14 days. Perfusion was restored during the first 7 days following surgery and plateaued by day 14 (*n* = 5 animals). *C*, representative H&E-stained brightfield images of collateral arteries within non-ischaemic and ischaemic adductors at Day 7 after surgery. Collateral arteries were identified by the neurovascular bundle which contained (N) nerve, (A) artery and (V) vein. The neurovascular bundle was imaged at 20× and at 40× magnification with an Olympus BX51 microscope. *D*, representative epifluorescence immunostaining of collateral arteries (CD31+$\alpha$SMA+) and diameter measurements within the semimembranosus muscle of ligated and unligated adductors at Day 7 after HLI. Collateral arteries in ischaemic adductors expanded in response to HLI compared to non-ischaemic adductors. The semi-membranous muscle was imaged as it contains collateral vessels that are anatomically identifiable in the adductor muscle of the hindlimb. Collateral vessels from five separate animals were measured and used for statistical analysis (*n* = 5). Data are shown as means ± SEM and analysed using Student's *t* test with Welch's correction. *E*, gene expression changes of canonical Notch targets *Notch1*, *Dll4*, *Jag1*, *Hes1* and *Hey1* from sorted ischaemic arterial endothelial cells (Cd45−/Cd31+/Sca1+/Pdpn−) 5 days following HLI. Expression data were obtained by qRT-PCR and normalized to non-ischaemic contralateral limbs of each individual animal (*n* = 3 separate animals). Data are shown as means ± SEM and analysed using Student's *t* test with Welch's correction. *F*, sorting strategy and flow analysis of arterial endothelial population in ischaemic and contralateral non-ischaemic adductor muscles. Pan endothelial cells were identified based on cell surface staining of Cd45− and Cd31+. Arterial and lymphatic endothelial cells were further delineated based on Sca1+ and Pdpn+ staining. Haematopoietic cells were identified based on the lineage marker CD45+ and Ter119+. [Colour figure can be viewed at wileyonlinelibrary.com]

new targets of Notch signalling during postnatal vascular regeneration. We first used the murine model of hindlimb ischaemia using 8−12 week old C57Bl/6 wild-type (WT) mice in which unilateral hindlimb ischaemia (HLI) was induced by excision of the superficial femoral artery. Side branches were cauterized to shunt blood flow to collateral arteries found in the adductor muscle (Fig. 1*A*). Perfusion of hindlimbs was measured over the course of 14 days by laser Doppler to relate blood flow to vascularization (Fig. 1*B*). Time-dependent recovery was seen within the first 7 days following HLI, which gradually plateaued by Day 14 (Fig. 1*B*). Immunohistochemistry of collateral vessels contained within the adductor muscle 7 days after HLI showed and ∼2-fold increase in collateral vessel diameter compared to control contralateral limbs (Fig. 1*C* and *D*).

To determine changes in Notch signalling following HLI, arterial endothelial cells (aECs) were sorted from adductor muscles 5 days after HLI based on arterial specific cell surface markers (Cd45−/Cd31+/Sca1+/Pdpn−) (Xu *et al.* 2018). qRT-PCR of aECs showed significant upregulation of Notch signalling components, the receptor *Notch1*, ligand *Jag1* and downstream target *Hes1* (Fig. 1*E*). Although many of the cells present in ischaemic adductors were of haematopoietic origin as they were CD45+, aECs comprising a fraction of the endothelial population were recovered similarly to non-ischaemic tissue on Day 5 following ischaemia onset (Fig. 1*F*). These data showed upregulation of Notch signalling in aECs before recovery of perfusion.

## Inhibition of Notch signalling delays vascular regeneration *in vivo*

To address the role of endothelial Notch signalling in vascular regeneration, we used the loss-of-function genetic system in which expression of DNMAML prevents recruitment of essential co-factors to the core transcriptional complex composed of RBPJ, NICD and MAML; it thereby acts as highly specific transcriptional repressor of Notch signalling (Tu *et al.* 2005). The DNMAML-GFP transgene is flanked by a transcriptional stop cassette at the Rosa26 locus, which enabled endothelial-specific inhibition of Notch signalling upon crossing DNMAML-GFP mice with the inducible VE-cadherin-Cre (Cdh5-CreERT2) strain (Sörensen *et al.* 2009). Inhibition of Notch signalling in endothelial cells was evident after four daily tamoxifen injections followed by a week-long rest period to allow tissue-specific expression of DNMAML (Fig. 2*A*). Loss of Notch signalling through endothelial cell-specific expression of DNMAML was validated by measuring protein accumulation of the downstream target Hes1 as well as GFP-tagged DNMAML in the sorted endothelial cells from DNMAML-Cdh5-CreERT2 Cre-positive mice compared to Cre-negative littermates (Fig. 2*B*).

DNMAML mice showed defective revascularization on Day 7 following HLI compared to Cre-negative littermates (Fig. 3*A* and *B*). Immunohistochemical staining of Cd31+/$\alpha$-Sma collateral arteries in the adductor muscle from the ischaemic and non-ischaemic contralateral limb

in DNMAML mice showed no significant expansion following ischaemia (Fig. 3*C* and *D*).

## Activation of Notch signalling reverses the impaired perfusion phenotype

We next determined whether enhancing Notch activity in endothelial cells would promote functional recovery of perfusion following HLI. Notch signalling was augmented through an *in vivo* gain-of-function genetic system by inducible over-expression of NICD (Murtaugh *et al.* 2003). NICD expression was restricted to endothelial cells through the inducible endothelial specific driver, VE-cadherin-Cre (Cdh5-CreERT2) (Fig. 4*A*). Although DNMAML and NICD mice had similar perfusion levels following HLI, NICD mice showed significantly improved limb perfusion on Days 5 and 7 compared to control DNMAML mice (Fig. 4*B* and *C*). Enhanced perfusion in NICD mice may in part be the result of arterio-genesis as vessel lumens of collateral arteries significantly expanded by Day 7 following HLI, in contrast to arteries of DNMAML mice (Fig. 4*D* and *E*). Expression of the Notch ligand *Dll4* and downstream target *Hey1* were significantly greater in sorted aECs of NICD adductors as compared to aECs in DNMAML adductors (Fig. 4*F*). Additionally, aECs sorted from DNMAML adductors expectedly showed reduced expression of the canonical Notch targets *Notch1*, *Jag1* and *Hes1* as compared to

WT controls (Fig. 4*F*). Moreover, the phenotypic aECs in ischaemic adductors were fully restored by Day 7 in WT and NICD mice, in contrast to DNMAML mice, which contained significantly fewer aECs in ischaemic adductors (Fig. 4*G*). Arterioles identified based on immunohistochemical staining of CD31+/αSMA+ vessels showed no differences in the number in ischaemic adductors of WT, DNMAML and NICD mice (Fig. 4*H*). Thus, the results together showed Notch signalling is a key mechanism of aEC generation and may mediate adaptive revascularization through microvascular expansion.

## Downstream targets of Notch signalling identified by RNAseq

To delineate the pathways downstream of Notch in controlling arterial regeneration and perfusion recovery, aECs from ischaemic and non-ischaemic adductors from WT, DNMAML and NICD mice were sorted 5 days after HLI for bulk RNAseq. WT reporter mice contained a loxP-flanked transcription stop site, which prevented the expression of tdTomato fluorescent protein (Madisen *et al.* 2010). When crossed with the same inducible endothelial Cre-specific driver and following Cre induction, all endothelial cells were labelled. aECs were sorted on the basis of the aEC markers CD31+/Sca1+/PDPN-/Cd45−. aECs from aorta and

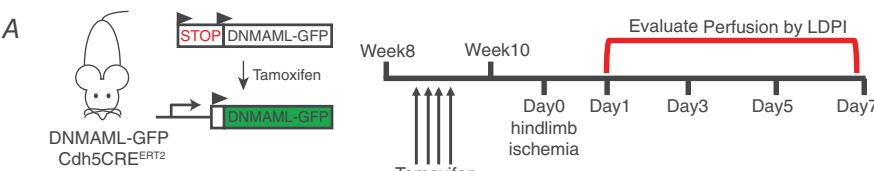

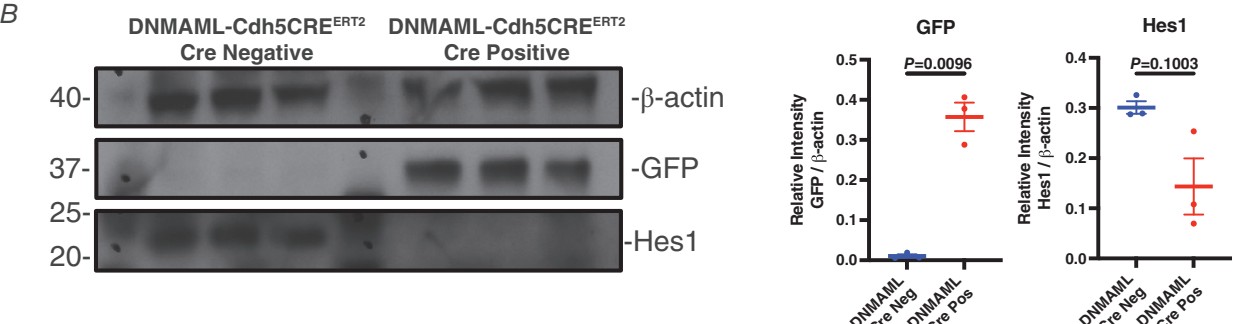

**Figure 2. Inhibition of endothelial Notch signalling via DNMAML overexpression**
*A*, schematic overview of hindlimb ischaemia experiments in DNMAML-Cdh5CRE[ERT2] Cre-negative and DNMAML-Cdh5CRE[ERT2] Cre-positive mice. Four rounds of daily TAM injections were administered in 8-week-old mice. Ischaemia and LDPI were performed 2 weeks after the first TAM injection. *B*, inhibition of Notch signalling in endothelial cells was confirmed by immunoblot of Hes1 (30 kDa) and GFP (35 kDa) from endothelial cells purified from lungs of DNMAML-Cdh5CRE[ERT2]-positive and -negative littermates. Transgenic expression of GFP was restricted to endothelial cells in DNMAML-Cdh5CRE[ERT2] mice. Cre-negative mice did not express GFP. Immunoblot analysis was conducted by comparing the relative optical density of proteins of interest, GFP or Hes1, to *β*-actin. [Colour figure can be viewed at wileyonlinelibrary.com]

lungs were used as gating controls. Isolation purity of endothelial cells was assessed by probing for GFP-tagged DNMAML and NICD (Fig. 5*A* and *B*). RNAseq data from all three genotypes were distinguishable by PCA (Fig. 5*C*). Statistical analysis by one-way ANOVA identified 6883 genes differentially expressed among the three groups (Fig. 5*D*). To narrow the number of direct target genes, we compared differentially expressed genes from the Notch inhibition group (DNMAML) and Notch overexpression group (NICD); here we identified 3438 differentially expressed genes (Fig. 6*A*). We also confirmed the purity of aECs based on the upregulation of arterial genes *Hey1*, *Dll4*, *Notch1* and *Efbn2* in the NICD group (Fig. 6*A*). Moreover, we confirmed that the previously established Notch targets *Hes1*, *Hey1* and *Dtx1* were downregulated predictably in the DNMAML group

(Fig. 6*A*). Gene set enrichment and pathway analysis showed upregulation of fibrosis, granulocyte adhesion and atherosclerosis signalling pathways secondary to activation of inflammatory conditions following blood flow stoppage (Fig. 6*B*).

Interestingly, we identified *Sema3g* satisfying the criteria of a Notch target based on its gene expression being higher in NICD and WT groups than the DNMAML group. Moreover, Sema*3g* expression was 40-fold greater in NICD than DNMAML groups (Fig. 6*A*). Ingenuity pathway analysis placed *Sema3g* as an axonal guidance cue (Fig. 6*B*). Alongside *Sema3g*, other established Notch targets *Efbn2* and *Unc5b* were also identified (Fig. 6*C*). Although *Sema3g* was the most upregulated Class 3 Semaphorin, *Sema3b*, *Sema3c*, *Sema3d* and *Sema3e* were also upregulated to varying

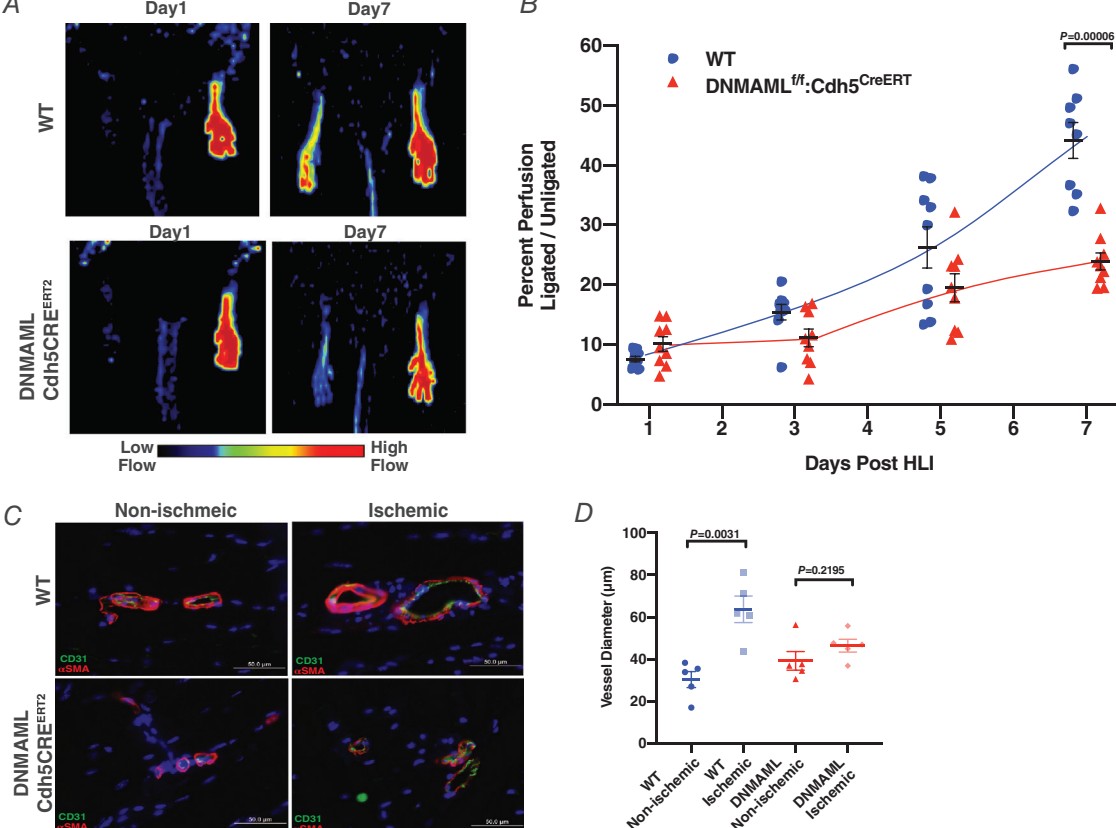

**Figure 3. DNMAML delays perfusion recovery and prevents revascularization**
*A*, representative laser Doppler images of WT and DNMAML-Cdh5CRE^ERT2 mice at Day 1 and Day 7 following hindlimb ischaemia. *B*, perfusion measurements of WT and DNMAML-Cdh5CRE^ERT2 hindlimbs measured over the course of 7 days after ischaemia. Perfusion of DNMAML-Cdh5CRE^ERT2 mice was compared to WT controls on Days 1, 3, 5 and 7 by Student's *t* test with the Holm–Sidak method (*n* = 9 animals). *C*, representative epifluorescence immunostaining of collateral arteries (CD31+/αSMA+) within non-ischaemic and ischaemic adductors of WT and DNMAML-Cdh5CRE^ERT2 mice 7 days after ischaemia. *D*, diameter measurements of collateral arteries (CD31+/αSMA+) within the neurovascular bundle of ligated adductors and un-ligated contralateral adductors. The neurovascular bundle within the semi-membranous muscle was imaged because it is anatomically identifiable in the adductor muscle of the hindlimb. Collateral vessels from five separate animals were measured and used for statistical analysis (*n* = 5). Data are shown as means ± SEM and analysed using Student's *t* test with Welch's correction. [Colour figure can be viewed at wileyonlinelibrary.com]

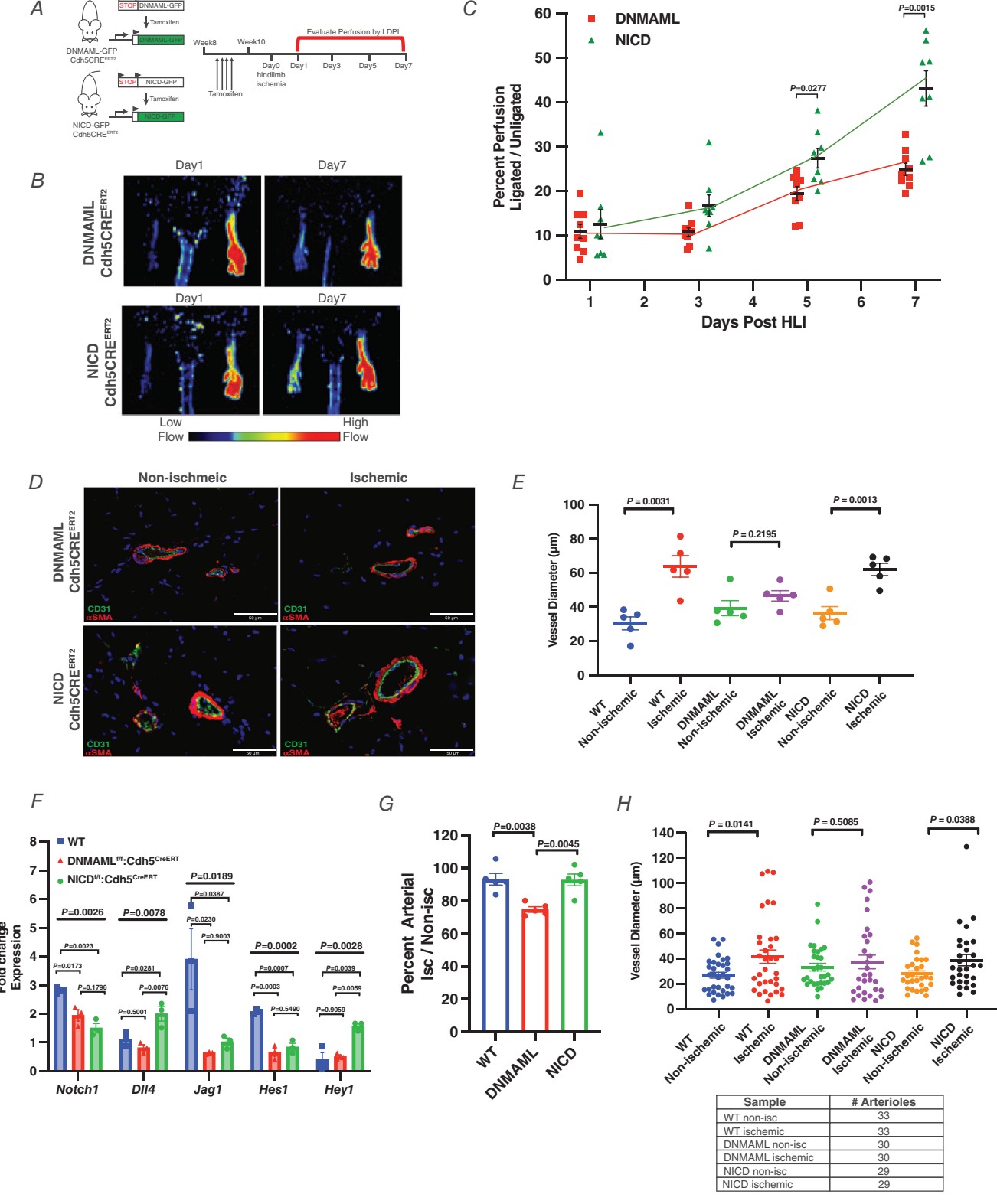

**Figure 4. Augmented endothelial Notch signalling improves perfusion recovery following hindlimb ischaemia**

*A*, schematic overview of hindlimb ischaemia experiments in DNMAML-Cdh5CRE^ERT2 and NICD-Cdh5CRE^ERT2 mice. Four rounds of daily TAM injections were administered in 8-week-old mice. Ischaemia and LDPI were performed 2 weeks after the first TAM injection. *B*, representative laser Doppler images of DNMAML-Cdh5CRE^ERT2

and NICD-GFP: Cdh5CRE$^{ERT2}$ mice at Day 1 and Day 7 following hindlimb ischaemia. *C*, perfusion measurements of DNMAML-Cdh5CRE$^{ERT2}$ and NICD-GFP-Cdh5CRE$^{ERT2}$ hindlimbs measured over the course of 7 days after ischemia. Perfusion of NICD-GFP-Cdh5CRE$^{ERT2}$ mice was compared to DNMAML mice at Days 1, 3, 5 and 7 by Student's *t* test with the Holm–Sidak method (*n* = 8 animals). *D*, representative epifluorescence immunostaining of (CD31+/αSMA+) collateral arteries within non-ischaemic and ischaemic adductors of DNMAML-Cdh5CRE$^{ERT2}$ and NICD-GFP: Cdh5CRE$^{ERT2}$ mice taken 7 days after ischaemia. *E*, diameter of WT, DNMAML and NICD collateral arteries (CD31+/αSMA+) from ligated adductors and un-ligated contralateral adductors. Diameter of collateral arteries increased in ischaemic adductors of NICD but not DNMAML mice following hindlimb ischaemia. WT and DNMAML collateral artery measurements were obtained from previous experiments shown in Figs 1 and 3*D*. Collateral vessels from five separate NICD animals were measured and used for statistical analysis (*n* = 5). Data are shown as means ± SEM and analysed using Student's *t* test with Welch's correction. *F*, gene expression changes of canonical Notch targets *Notch1*, *Dll4*, *Jag1*, *Hes1* and *Hey1* of sorted ischaemic arterial endothelial cells (Cd45−/Cd31+/Sca1+/Pdpn−) 5 days following hindlimb ischaemia. Expression data for WT, DNMAML and NICD mice were obtained by qRT-PCR and normalized to non-ischaemic contralateral limbs of each individual animal (*n* = 3 separate animals). WT qPCR results were obtained from previous experiments shown in Fig. 1*F*. Data are shown as means ± SEM and analysed using ordinary one-way ANOVA with Tukey's multiple comparisons test. *G*, recovery analysis of arterial endothelial cells population (Cd45−Cd31+Sca1+Pdpn−) in ischaemic adductors of NICD and WT compared to DNMAML mice. The arterial endothelial population recovered to a lesser extent in DNMAML adductors 5 days after ischaemia. Recovery was calculated based on the percentage of arterial endothelial cells found in ischaemic adductors compared to the non-ischaemic contralateral limb of each animal (*n* = 5 animals). Data are shown as means ± SEM and analysed using Student's *t* test with Welch's correction. *H*, vessel number and diameter were measured in ischaemic and contralateral control adductor muscle of each genotype. Arterioles were identified based on immunohistochemical staining of CD31+/αSMA+ vessels. Five separate animals were used for analysis. Data are shown as means ± SEM and analysed using Student's *t* test with Welch's correction [Colour figure can be viewed at wileyonlinelibrary.com]

degrees (Fig. 6*D*, right panel). From our analysis we also identified *Cxcl5* and *AplnR* as other candidate genes downstream of Notch signalling based on their expression pattern; *Cxcl5* was upregulated in the NICD group and downregulated in the DNMAML group (Fig. 6*A*). Conversely, expression of *AplnR* was inversely affected by the level of Notch signalling, as gene expression was higher in the DNMAML than NICD group (Fig. 6*D*, left panel).

To validate the transcriptomics analysis, we conducted qRT-PCR experiments of aECs isolated from ischaemic adductors of WT, DNMAML and NICD mice 5 days after HLI. *Sema3g* was upregulated in both NICD and WT cells whereas it was suppressed in DNMAML. Although the expression of *Cxcl5* and *Aplnr* were different in the WT, DNMAML and NICD groups, the qRT-PCR expression profile of aECs did not align with our initial RNAseq analysis (Fig. 6*E*). To address *Sema3g*, *Cxcl5* and *Aplnr* as potential Notch targets, HPAECs were treated with GSI, a pan inhibitor of Notch signalling (Cook et al. 2012). *Sema3g* expression was markedly reduced, comparable to the positive control *Hes1*, a canonical Notch target. Removal of GSI from HPAECs by washout restored the expression of *Hes1* and *Sema3g* (Fig. 6*F*). Although *Cxcl5* expression was reduced by 50% with GSI, it was inhibited to a lesser degree than *Sema3g* and *Hes1* (Fig. 6*F*). *Aplnr* expression was enhanced upon treatment with GSI but was unchanged after GSI removal. While *Cxcl5* and *Aplnr* supported our RNAseq results, only *Sema3g* exhibited many features of a direct downstream target of Notch signalling in endothelial cells; thus, the role of Sema3g was addressed further.

## Sema3g inhibits neovascularization

As our results showed that Notch signalling was critical in regulating revascularization, we next addressed the possible role of Sema3g in this response. Here we used two relevant models to address Sema3g function. First, we utilized an established endothelial spheroid assay in which vessels sprout radially to form neovessels (Korff & Augustin, 1998). HPAECs were transduced to overexpress DNMAML or GFP, as a negative control (Fig. 7*A*). DNMAML- and GFP-expressing HPAEC spheroids were formed and embedded in 3D collagen gels containing VEGF$_{165}$ and FGF, which resulted in sprout formation within 24 h (Fig. 7*B*). HPAECs expressing DNMAML produced significantly more sprouts covering a greater surface area than the control (Fig. 7*B* and *C*), consistent with response to the loss of Notch signalling in augmenting VEGF signalling in endothelial cells and promoting the aberrant hyper-sprouting phenotype (Jakobsson *et al.* 2010).

We next addressed the possibility that Sema3g functions as a negative regulator of VEGF/Notch signalling and thus controls vascular patterning though its receptors Nrp2 and PlexinD1 (Gaur *et al.* 2009). *SEMA3G* was expressed in HEK293T cells, and conditioned medium containing SEMA3G (Fig. 7*D*) was added to GFP-expressing control spheroids. The addition of SEMA3G inhibited sprout formation and covered less surface area than vehicle-treated spheroids (Fig. 7*B* and *C*). DNMAML-expressing spheroids responded to SEMA3G treatment in a similar manner by generating fewer sprouts and covering less area than vehicle-treated control spheroids (Fig. 7*B* and *C*). Thus, SEMA3G

functioned to regulate sprout formation and the spread of neovessels.

To further dissect Sema3g signalling, we performed qRT-PCR screening of the cognate Semaphorin receptors, Nrp and Plexin, using confluent GFP-expressing HPAEC controls and HPAECs expressing DNMAML. Expression of *Nrp1* and *Nrp2* was elevated, while the putative Notch targets *Notch1, Hes1* and *Sema3g* were reduced in DNMAML-expressing cells compared to

control (Fig. 8*A*). *PlexinD1*, a co-receptor required for Semaphorin signalling, was also downregulated in DNMAML-expressing HPAECs compared to controls (Fig. 8*A*). These findings together suggest that Sema3g is a downstream target of Notch that regulates VEGF signalling through neuropilin receptors known to selectively bind to VEGF isoforms as well as Class 3 Semaphorins (Sunyoung *et al.* 2007). We demonstrated that $VEGF_{165}$ treatment of DNMAML HPAECs and

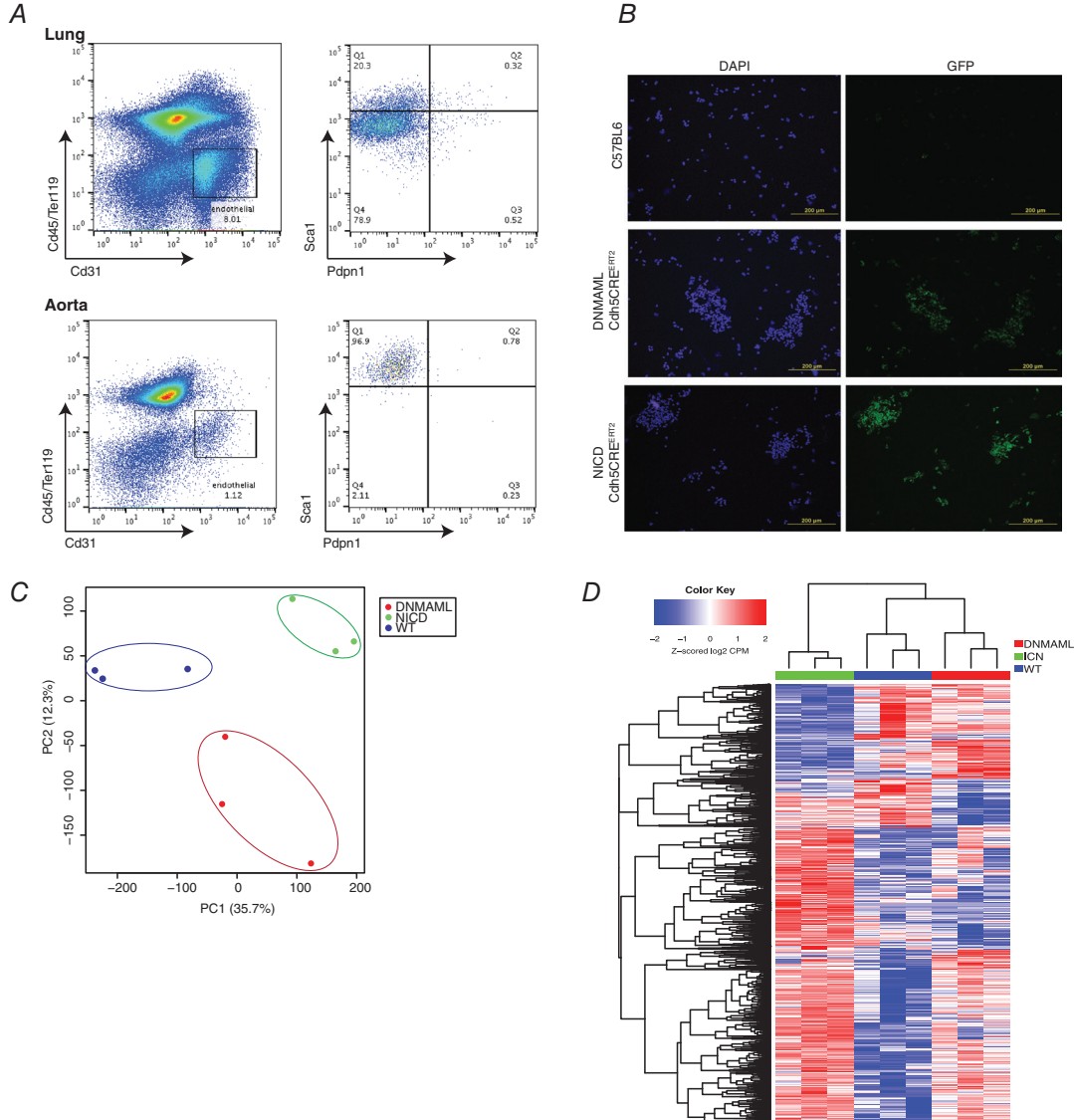

**Figure 5. RNAseq of DNMAML-Cdh5CRE$^{ERT2}$ and NICD-Cdh5CRE$^{ERT2}$ arterial endothelial cells following hindlimb ischaemia**

*A*, representative flow plots used to sort Cd45−/Ter119−/CD31+/Sca1$^{high}$/Pdpn1− endothelial cells from ischaemic contralateral adductors. Endothelial cells from the aorta and lung were used as gating controls. *B*, purity of sorted arterial endothelial cells used in RNAseq experiments was verified based on GFP staining. Transgenic Notch mutant DNMAML-Cdh5CRE$^{ERT2}$ and NICD-Cdh5CRE$^{ERT2}$ mice express a GFP reporter, while C57BL6J do not. *C*, principal component analysis (PCA) analysis of WT, DNMAML and NICD ischaemic arterial endothelial cells used for transcriptomics analysis ($n = 3$ animals from each group). *D*, differential gene expression analysis of WT, DNMAML and NICD ischaemic groups. One-way ANOVA was performed across all three genotypes identifying 6883 differentially expressed genes. [Colour figure can be viewed at wileyonlinelibrary.com]

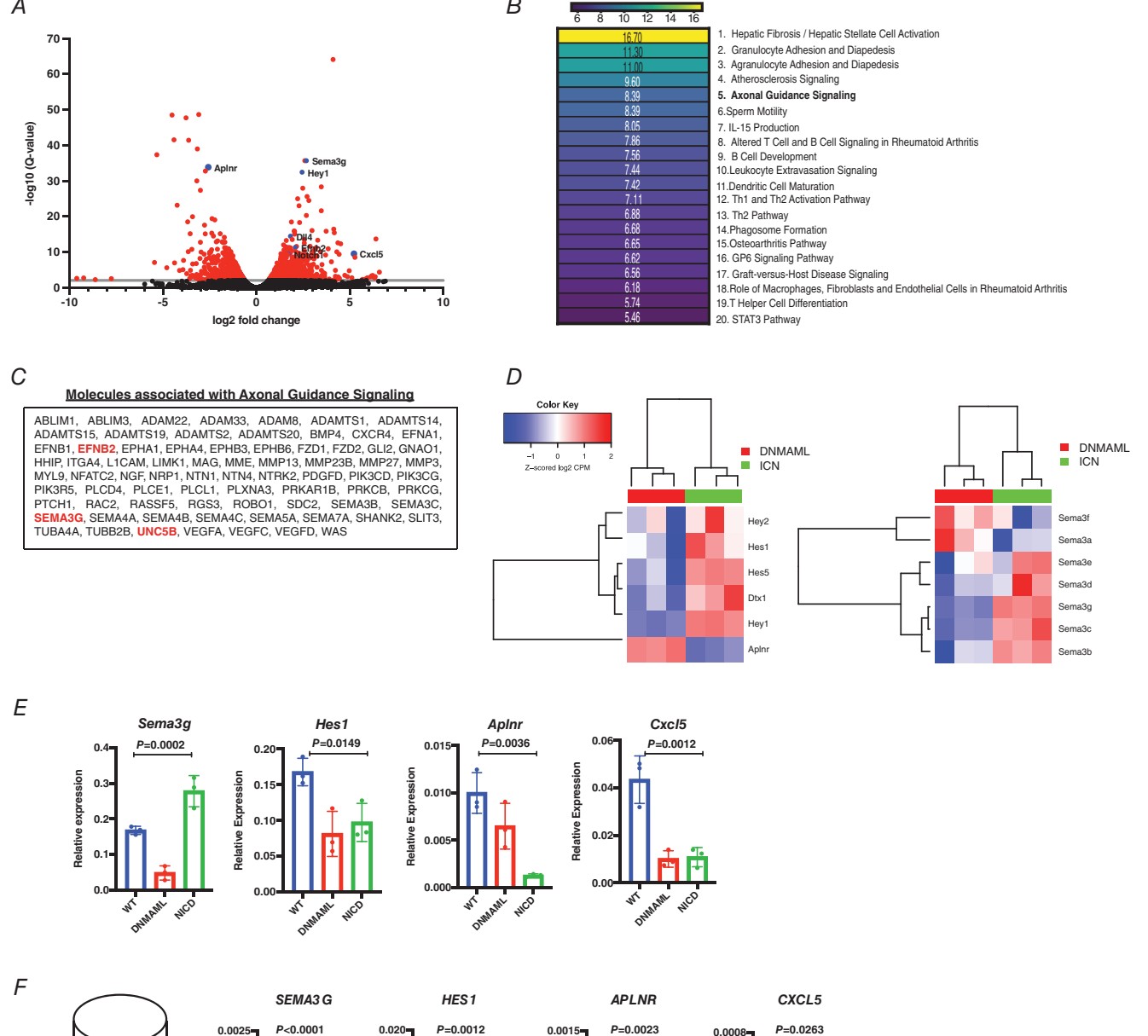

**Figure 6. Sema3g as a target of Notch signalling in arterial endothelial cells**
*A*, to identify Notch-dependent genes the transcriptome of DNMAML and NICD ischaemic arterial endothelial cells was compared. A volcano plot of 3438 differentially expressed genes show upregulation of known arterial genes *Notch1*, *Efnb2*, *Dll4* and *Hey1*. *Sema3g* was found to be highly upregulated in the NICD group compared to DNMAML group. Genes with a false discovery rate (FDR) value <0.01 are highlighted in red. *B*, list of the top20 signalling pathways enriched following ingenuity pathway analysis of differentially expressed genes in DNMAML-Cdh5CRE^ERT2 and NICD-Cdh5CRE^ERT2. arterial endothelial cells. Axonal guidance signalling is highlighted as this pathway contained potentially new as well as previously validated Notch targets. *C*, list of genes identified to be involved in axonal guidance signalling from the ingenuity pathway analysis of differentially expressed genes in DNMAML-Cdh5CRE^ERT2 and NICD-Cdh5CRE^ERT2 arterial endothelial cells. Potential Notch targets are highlighted in red. *D*, Z-score heatmap of canonical endothelial Notch targets and all Class3

Semaphorins were analysed in DNMAML, ICN and WT ischaemic groups. Expression of Notch targets *Dtx1*, *Hey1*, *Hes5*, *Hes1* and *Hey2* was upregulated in NICD compared to DNMAML. Corresponding to Notch levels, expression of *Sema3b*, *Sema3c*, *Sema3d*, *Sema3e* and *Sema3g* are upregulated in NICD compared to DNMAML. *E*, qRT-PCR validation of *Sema3g*, *Hes1*, *Cxcl5* and *Aplnr* expression in sorted arterial endothelial cells from the indicated transgenic mice. Data are shown as means ± SEM and analysed using one-way ANOVA (*n* = 3 animals from each group). *F*, schematic describing gamma secretase inhibitor (GSI) and washout (WO) experiments. qRT-PCR expression of *Sema3g*, *Hes1* and *Cxcl5* was inhibited under GSI but recovered following an 8 h WO (*n* = 3 separate experiments). Data are shown as means ± SEM and analysed using one-way ANOVA. [Colour figure can be viewed at wileyonlinelibrary.com]

control HPAECs strongly activated the VEGF signalling pathway as evident by the downstream phosphorylation of Tyr1175 of VEGFR2 (Fig. 8*B*). Pre-treatment with SEMA3G reduced VEGF$_{165}$-induced phosphorylation of VEGFR2 (Fig. 8*B*). Thus, SEMA3G negatively regulated VEGF-mediated Notch signalling that may contribute to orderly vascularization and vascular network formation. Together, our results reveal a signalling mechanism in which Semaphorins act as a negative feedback regulator of Notch signalling to limit tip cell generation and arrange an ordered vascular network.

To test the application of Sema3g-mediated down-regulation of VEGF/Notch signalling in promoting orderly vascular regeneration, we carried out studies using whole mount retinas in DNMAML and WT littermates on postnatal Day 6 (P6), a well-established method for assessment of vascular patterning (Fig. 9*A*). In this model, retinal vasculature begins at birth forming an initial vascular plexus, and endothelial cells migrate outwards establishing a highly organized vascular tree composed of arteries and veins by postnatal P7. DNMAML-Cdh5-CreERT2 mice, as used in previous studies, were injected daily with tamoxifen at P1–P3 to inhibit Notch signalling in endothelial cells. Retinas were harvested at P6 and stained with isolectinB4 to visualize vascular network formation and angiogenic fronts. Retinas from DNMAML mice exhibited compacted angiogenic fronts with aberrantly increased vascular density and greater number of filopodia compared to control (Fig. 9*B* and *C*). These results, corresponding with the above *in vitro* data (Fig. 4*A* and *B*), demonstrated that loss of Notch signalling leads to an unruly hyper-sprouting phenotype due to the dysregulated function of tip and stalk cells (Hellström *et al.* 2007).

We next tested whether SEMA3G administration *in vivo* in this system would resolve pathological vascular network formation observed with inhibition of Notch signalling. Following tamoxifen injections at P1–P3, recombinant SEMA3G (1 μg) or vehicle was delivered intradermally near the jugular vein in two intervals at P4 and P5 (Fig. 9*D*). P6 retinas were then dissected and the vasculature was stained to visualize neovessel formation. SEMA3G had a profound effect in reducing vascular density at angiogenic fronts and filopodia formation in DNMAML retinas and preventing aberrant vascularization seen in controls (Fig. 9*E* and *F*). Thus,

Notch signalling coordinates vascular pattering through Sem3g, which in turn functions by modulating VEGF signalling to induce proper patterning to the growing vasculature.

## Discussion

Several adaptive responses are activated following vascular occlusion with the physiological intent to restore blood flow. The key mechanism of revascularization and vascular development depends on VEGF. The arrest of blood flow in capillaries is followed by secretion and establishment of the VEGF gradient, which is essential for new vessel formation (Limbourg *et al.* 2009). End-othelial cells migrate towards ischaemic VEGF, generating vessels in an organized manner by establishing a tip–stalk cell identity via activation of Notch signalling (Phng & Gerhardt, 2009). Furthermore, pre-existing collateral arteries, which divert blood from larger arteries to smaller arterial vessels, respond to ischaemia and undergo arteriogenesis (He *et al.* 2006). Shunting of blood from the occluded arteries causes collateral arteries to expand through proliferation of endothelial cells and recruitment of smooth muscle cells (Cooke & Losordo, 2015). The importance of Notch signalling in angiogenesis and revascularization has been addressed through studies of transcription factors activated by VEGF/Notch signalling (Hogan *et al.* 2017). Outside of *Ephb2*, which demarks arteries and veins (Adams *et al.* 1999), the downstream targets of Notch signalling mediating vascularization remain obscure. Here we used transgenic mice that either inhibited or hyperactivated Notch signalling to identify Notch-dependent targets in endothelial cells in models of hindlimb revascularization and retinal vascularization. The study of these disparate vascular beds enabled us to make generalizable conclusions about the role of Notch activation of downstream pathways essential for productive vascularization in distinct vascular beds.

In hindlimb studies, we used a standard model of ischaemia produced by ligation of the femoral artery (Limbourg *et al.* 2009). Its advantage lies in quantitatively assessing the time course from loss to eventual re-establishment of vascularization and perfusion. We focused our search on Notch gene targets altered during loss of Notch or enhanced Notch conditions. Our parsing criteria classified direct Notch transcriptional targets

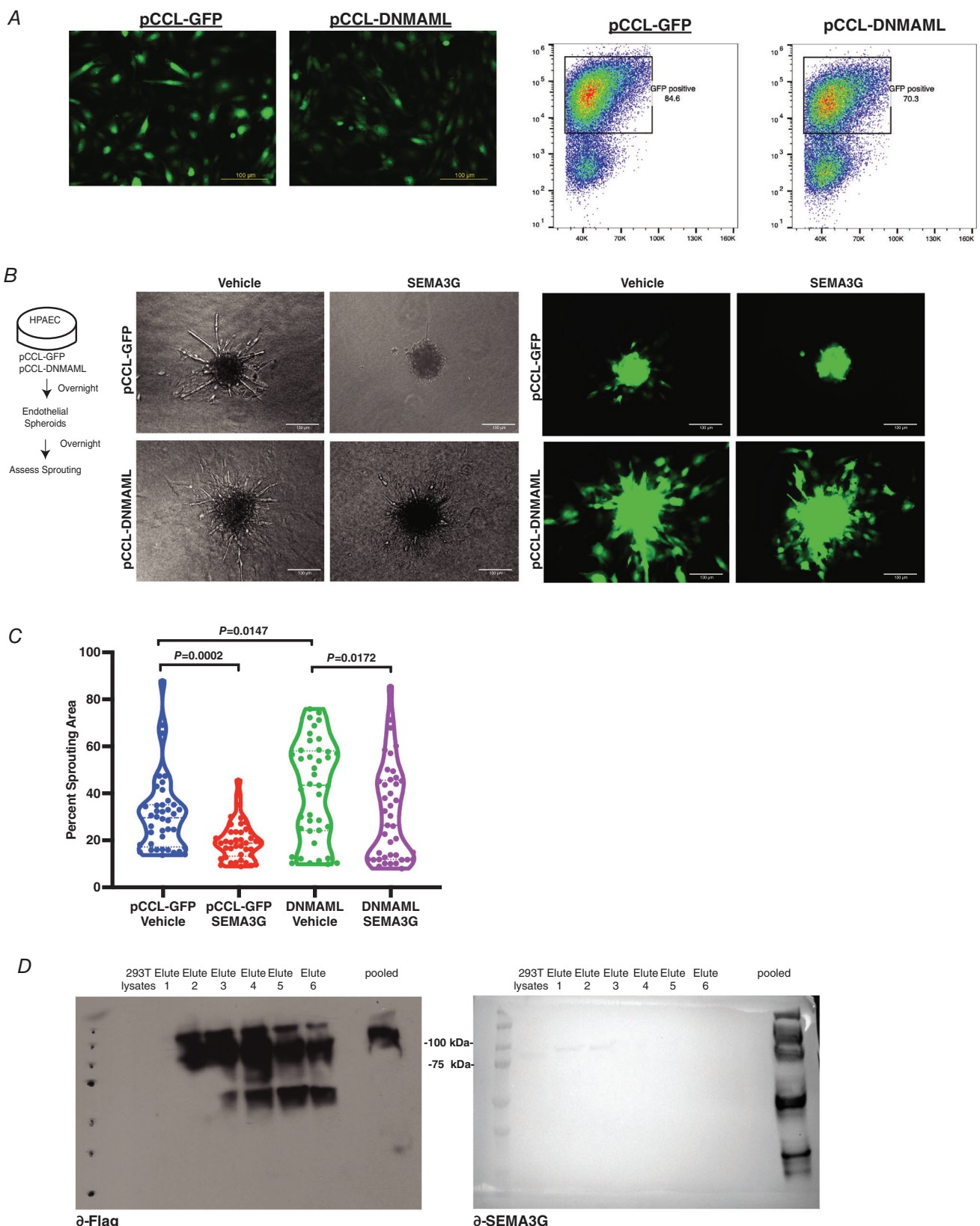

**Figure 7. SEMA3G rescues defective vessel sprouting formation in endothelial spheroid assay**
*A*, epifluorescence images and flow cytometry plots validating the transduction of human pulmonary arterial endothelial cells with GFP alone or DNMAML-GFP lentivirus. *B*, representative images of pCCL-GFP and

pCCL-DNMAML expressing HPAEC spheroids. Spheroids were embedded in collagen gels and treated with vehicle or SEMA3G overnight. Spheroids were selected at random and imaged in DIC and fluorescent channels to assess sprouting area. C, sprouting areas of pCCL-GFP and pCCL-DNMAML spheroids were compared with vehicle treatment or medium containing SEMA3G. Sprouting area was determined by the GFP signal expressed by endothelial cells and shown as a percentage in relation to the field of view (*n* = 40 spheroids). Data are shown as means ± SEM and analysed using Student's *t* test with Welch's correction. D, immunoblot of FLAG-purified recombinant SEMA3G used in *in vitro* angiogenesis assays. Individual elutions and pooled elutions (2–6) of purified SEMA3G were probed with αFLAG antibody and αSEMA3G. [Colour figure can be viewed at wileyonlinelibrary.com]

*Hes1* and *Hey1*, which were highly expressed in NICD (gain-of-function) and downregulated in DNMAML (loss of function) groups. Canonical Notch targets *Efnb2*, *Dll4* and *Notch1* were also found to be altered, reinforcing our confidence in this approach. Interestingly, we identified *Sema3g*, a class 3 Semaphorin, as a potential downstream Notch target regulating vascular regeneration and orderly vascular patterning.

We focused on *Sema3g* as it its expression was restricted to arterial cells, which play a key role in re-vacsularization (Kutschera *et al.* 2011). Originally described as secretory products of neurons, the Semaphorin class of peptides bind to Plexin and Nrp family receptors and cause rearrangement of the actin cytoskeleton and focal adhesions, regulating cell shape and motility (Neufeld & Kessler, 2008; Adams & Eichmann, 2010). They have

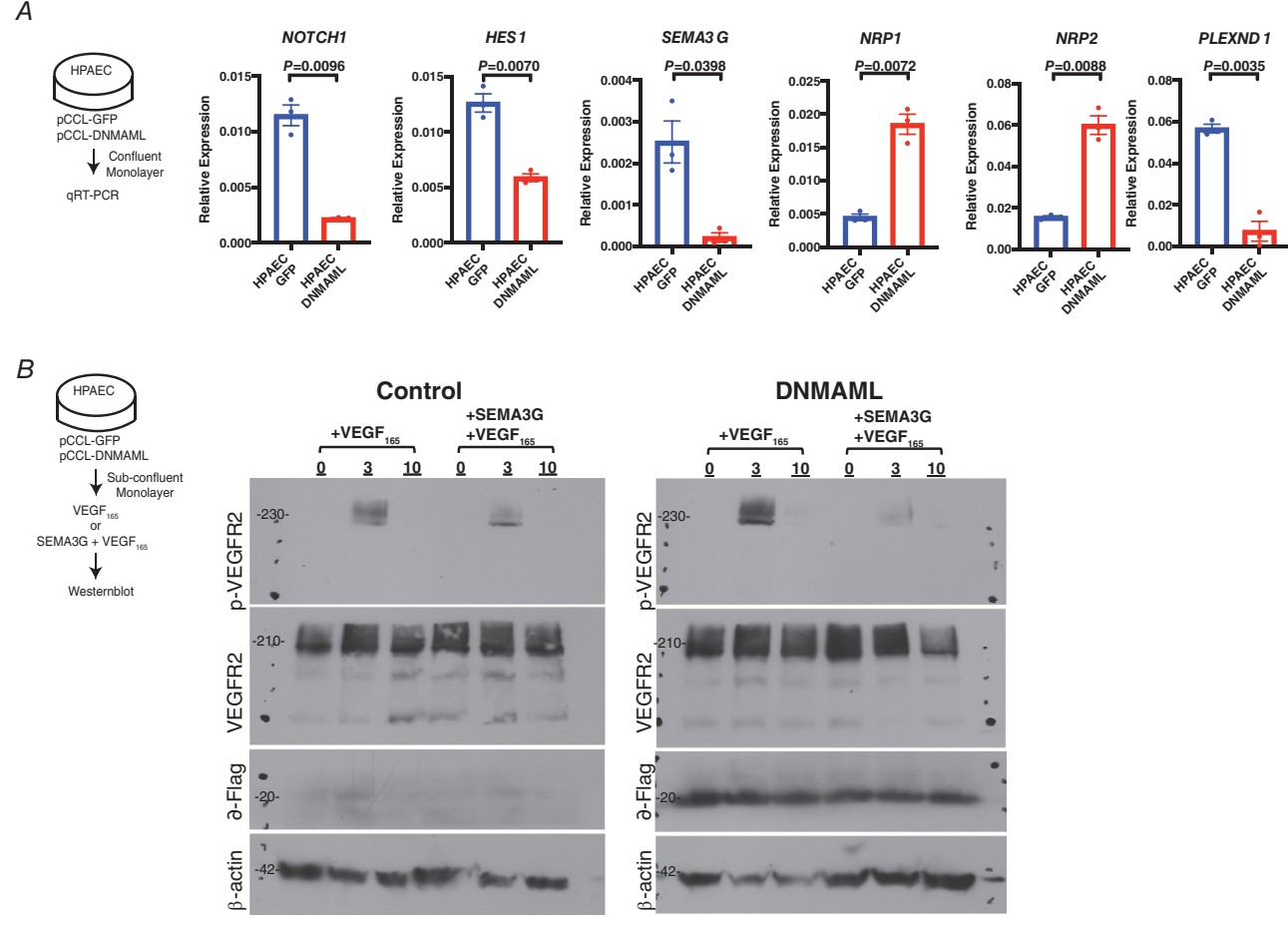

**Figure 8. SEMA3G is a negative feedback regulator of Notch signalling by attenuating VEGF signalling**
A, relative gene expression changes of *NOTCH1*, *HES1*, *SEMA3G*, *NRP1*, *NRP2* and *PLEXIND1* in pCCL-GFP and pCCL-DNMAML expressing HPAECs. Data are shown as means ± SEM and analysed using Student's *t* test with Welch's correction (*n* = 3 separate experiments). B, representative blot of VEGFR2 Tyr1175 phosphorylation in DNMAML expressing HPAECs and GFP expressing control HPAECs. VEGF stimulation was assessed by comparing levels of phospho-VEGFR2 to total VEGFR2 at 3 and 10 min following the addition of VEGF$_{165}$. To competitively block VEGF signalling, cells were pretreated with SEMA3G for 10 min before addition of VEGF$_{165}$. DNMAML expression was validated based on immunoblotting of flag protein (*n* = 3 separate experiments). [Colour figure can be viewed at wileyonlinelibrary.com]

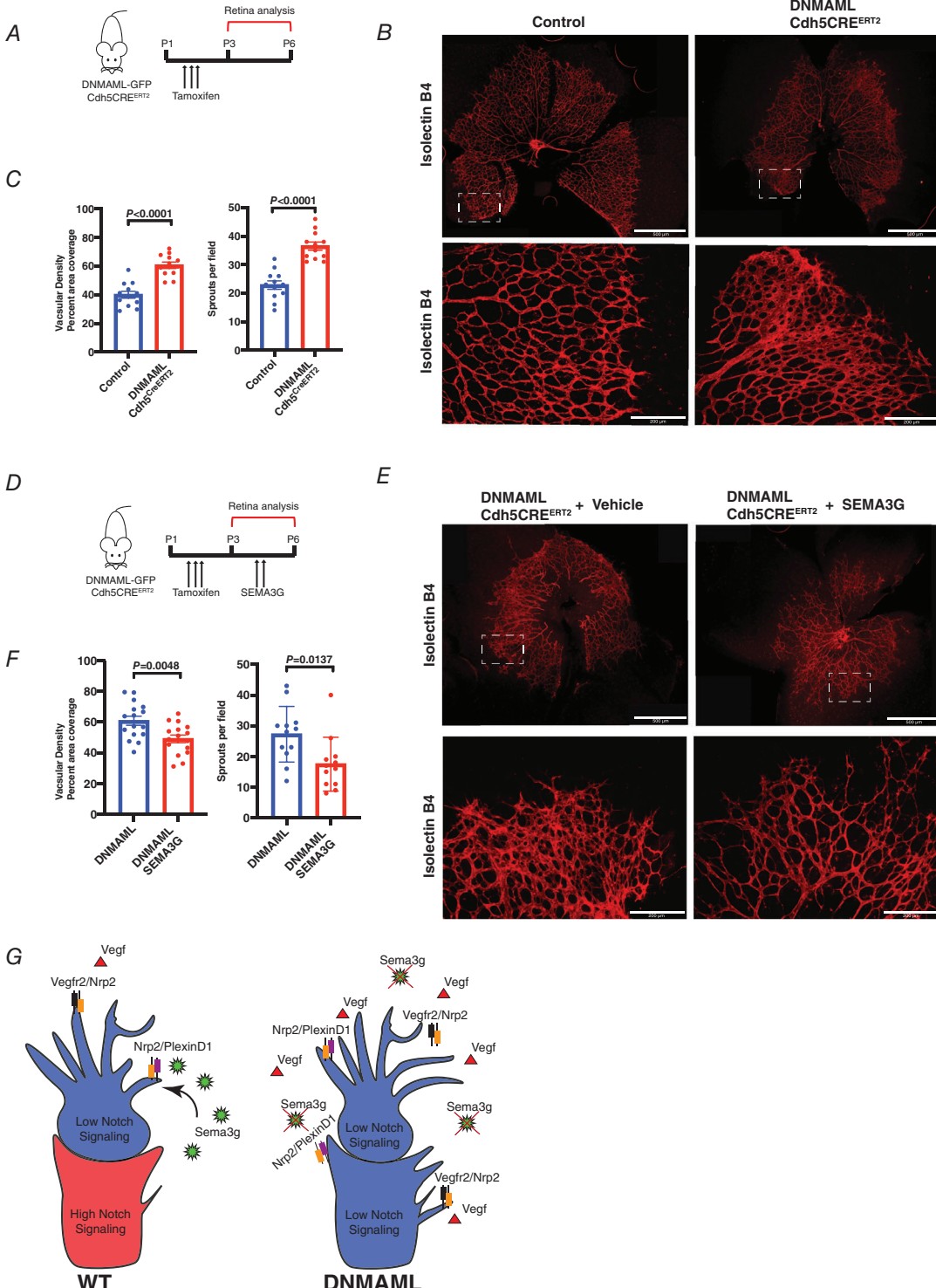

**Figure 9. SEMA3G normalizes vascular patterning in retinas of DNMAML-Cdh5CRE^ERT2 mice**
*A*, schematic of retinal angiogenesis model. Tamoxifen was injected in newborn control (Cre-) and DNMAML-Cdh5CRE^ERT2 mice (P1–P3) and retinas were harvested on P6. *B*, representative images of whole mount retinas (P6) and higher magnification images of angiogenic fronts of control (Cre-) and DNMAML-Cdh5CRE^ERT2 mice. Retinal vasculature was stained with isolectinB4 and is seen in red. *C*, vascular density and number of sprouts for control (Cre-) and DNMAML-Cdh5CRE^ERT2 mice at angiogenic fronts were quantified and compared. Vascular

density was determined based on the percentage of red signal as stained by isolectinB4 ells. Number of sprouts was calculated based on the number of filopodia present in each field of view. For analysis, four fields of view containing a vascular plexus were chosen for each retina. Data are shown as means ± SEM and analysed using Student's *t* test with Welch's correction (*n* = 3 separate animals). *D*, schematic of retinal angiogenesis experiments. Tamoxifen was injected in newborn DNMAML-Cdh5CRE$^{ERT2}$ (P1–P3) mice intragastrically. Recombinant SEMA3G (1 μg) was injected daily intradermally at P4 and P5. P6 retinas were dissected and stained with isolectinB4 to assess vascular defects by image analysis. *E*, representative images of whole mount retinas (P6) and angiogenic fronts of DNMAML-Cdh5CRE$^{ERT2}$ mice treated with vehicle and SEMA3G. *F*, vascular density and sprout numbers of DNMAML-Cdh5CRE$^{ERT2}$ mice treated with vehicle or SEMA3G were quantified and compared. Vascular density was determined by the percentage of red signal as stained by isolectinB4. Number of sprouts was calculated by the number of filopodia present in each field of view. For analysis, four fields of view containing a vascular plexus were chosen for each retina. Data are shown as means ± SEM and analysed using Student's *t* test with Welch's correction (*n* = 3 separate animals). *G*, model of Sema3g-Nrp2/PlexinD1 signalling in arterial endothelial cells. Under homeostatic conditions, stalk cells secrete Sema3g and antagonize VEGF signalling in tip cells to retract filopodia and prevent anastomosis of nearby blood vessels. Inhibition of Notch signalling suppresses Sema3g secretion, enhancing VEGF-Nrp2/Vegfr2 signalling to stimulate tip cell selection and promote filopodia extensions. [Colour figure can be viewed at wileyonlinelibrary.com]

primarily been shown to regulate axon migration and cause cone collapse in neurons, but more recent studies showed that they also regulated endothelial tip cell behaviour and outgrowth of new blood vessels (Larrivée *et al.* 2009; Sakurai *et al.* 2012). We demonstrated in both cultured endothelial cell experiments and retinal vessels that SEMA3G had profound anti-angiogenic effects in models of disordered and excessive angiogenesis.

In the present study, we utilized largely genetic approaches to inhibit Notch signalling and induce the hypersprouting phenotype, where the development of new blood vessels was compacted and unevenly dispersed in mouse retinas as well as in cultured endothelial spheroids. Administration of SEMA3G showed a striking anti-angiogenic effect on aberrant retinal microvasculature in DNMAML mice through inhibiting tip cell formation and improving vascular patterning seen in these mutants. SEMA3G also rescued the hypersprouting defect seen in DNMAML-expressing cultured endothelial cells. Based on the inhibition of tip cells, Sema3g functioned as a negative regulator of Notch signalling, suggesting a feedback mechanism whereby Notch drives *Sema3g* expression to downregulate excessive sprouting angiogenesis leading to normalization of vascular patterning.

During vascularization, endothelial cells respond to VEGF and upregulate Dll4 and stimulate Notch signalling in neighbouring stalk cells. Tip cells express high Nrp1 levels and therefore are more susceptible to VEGF cues (Hellström *et al.* 2007; Roca & Adams, 2007; Siekmann *et al.* 2008; Benedito *et al.* 2012). Thus, it appears that in circumstances such as ischaemic loss of perfusion, Sema3g functioned as a tissue-organizing factor that limits the number of endothelial cells responding to pro-angiogenic factors and functions to establish an organized vascular network. In settings of impaired and excessive angiogenesis such as oxygen-induced retinopathy or tumour-associated misshapen vessels (Bielenberg *et al.* 2004; Chen *et al.* 2021), Sema3g may

also contribute to reduction of tip cells and re-booting the angiogenesis programme. Sema3g appears to act via Nrp1, a receptor shared between Sema3g and VEGF (Gaur *et al.* 2009). Thus, our studies demonstrated that addition of SEMA3G in endothelial cells reduced the phosphorylation of VEGFR2, which perturbs VEGF signalling of angiogenesis (Kawamura *et al.* 2008; Blanco & Gerhardt, 2013b).

Our results showed that *Sema3g* was not only regulated by Notch signalling, as evident in the GSI washout experiments, where removal of the Notch inhibitor did not result in full restoration of *Sema3g* expression. Recent evidence showed that Hif2α directly regulated *Sema3g* expression (Chen *et al.* 2021). Thus, Hif2α may be a necessary co-factor driving the full expression of *Sema3g*. We also did not address the additive role of other Class 3 Semaphorins. Our RNAseq results suggest that *Sema3b*, *Sema3c* and *Sema3e* are also influenced by Notch signalling and thus may be involved in vascular regeneration and remodelling. Studies will be needed to dissect the importance and context of each or combinations of Semaphorins in vascular regeneration and patterning.

In the present study we largely focused on endothelial cells. It is possible that Notch signalling in vascular smooth muscle cells (Boucher *et al.* 2012) and inflammatory cells (Christopoulos *et al.* 2021) may also contribute to vascular regeneration and the release of Sema3g. For example, the Notch ligand Jag1 has been associated with mobilization, survival and proliferation of endothelial progenitors to promote neovascularization (Kwon *et al.* 2008). Dll1 was demonstrated to regulate the maturation of macrophages from monocytes and foster an anti-inflammatory environment (Krishnasamy *et al.* 2017). The adductor muscle following HLI contained many immune cells, as evident by the increased number CD45+ cells (Fig. 1*F*). It is therefore possible that immune cells stimulate Notch signalling in endothelial cells and support the generation of Semaphorins. We

cannot rule out cross-talk among these cells in regulating vascular regeneration. Our findings imply that Sema3g is a fundamental Notch target and illustrate the potential cross-talk between neuronal peptides and the vascular system. Blood vessels and nerves develop in apposition to each other and are both highly branched. Our studies showcase an overlooked and underappreciated guidance molecule that regulates vascular patterning.

Additional supporting information can be found online in the Supporting Information section at the end of the HTML view of the article. Supporting information files available:

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

# Additional information

## Data availability statement

Normalized gene count and differential expression analysis data are available as supplementary information. The data used to generate the analysis are available upon request to the senior author.

## Competing interests

None.

## Author contributions

J.H., M.L., J.R., K.V.P. and A.B.M. designed experiments and analysed the data. J.H. and M.L. performed the experiments. J.H., J.R., K.V.P. and A.B.M. wrote the manuscript.

## Funding

This work was supported by the following grants from the National Institutes of Health: T32HL007829, P01HL077806 to A.B.M. and R01HL134971 to K.V.P. Bioinformatics analysis in the project described was performed by the University of Illinois Research Informatics Core, supported in part by the National Centre for Advancing Translational Sciences, National Institutes of Health, Grant UL1TR002003.

## Keywords

angiogenesis, arteriogenesis, ischaemic injury, notch signaling, vascular biology

## Supporting information

Additional supporting information can be found online in the Supporting Information section at the end of the HTML view of the article. Supporting information files available:

**Peer Review History**
**Statistical Summary Document**
**RNAseq DEG analysis**

## Translational perspective

In this study we utilized transgenic mice that modulated Notch signalling to uncover Notch-dependent targets involved in revascularization following hindlimb ischaemia. We utilized systems of loss-of-function and gain-of-function in Notch signalling as it has been demonstrated to be central to vascular development and angiogenesis. Transcriptomics analysis of arterial endothelial cells from ischaemic tissue highlighted the involvement of neuronal peptides and identified Sema3g as a potential downstream target. *In vitro* and *in vivo* experiments assessing angiogenesis demonstrated Sema3g's profound effect in inhibiting tip cell formation. We postulate that Sema3g functions as a local organizer of new vessels invading the ischaemic tissue and thus mediating proper hierarchical organization of the vasculature. Our mechanistic results suggest that Sema3g attenuates VEGF signalling by competing for the co-receptor Nrp1 and highlight the need for future studies that target specific aspects of angiogenesis such as vessel pruning. The contribution of neuronal peptides is often underappreciated in studies outside of neurons, and future work investigating other class3 Semaphorins and the context in which they influence endothelial behaviour should be considered.

