## [Peer Review History · The Journal of Physiology]

Notch1 Promotes Revascularization through Modulation of Downstream Vascular Patterning Signaling

James Hyun, Monica Y Lee, Jalees Rehman, Kostandin V Pajcini, and Asrar B Malik
DOI: 10.1113/JP282286

Corresponding author(s): Asrar Malik (abmalik@uic.edu)

Review Timeline:

Submission Date:	19-Feb-2021
Editorial Decision:	25-Mar-2021
Resubmission Received:	27-Aug-2021
Editorial Decision:	05-Oct-2021
Revision Received:	28-Oct-2021
Accepted:	29-Nov-2021

Senior Editor: Don Bers

Reviewing Editor: Livia Hool

Transaction Report:

Dear Dr Malik,

Re: JP-RP-2021-281541 "Notch1 Dependent Semaphorin3g Signaling Induces Vascular Patterning and Reestablishes Arterialization Following Ischemia" by James Hyun, Monica Y Lee, Jalees Rehman, Kostandin V Pajcini, and Asrar B Malik

Thank you for submitting your manuscript to The Journal of Physiology. It has been assessed by a Reviewing Editor and by 2 Referees and the reports are copied below.

Please let your co-authors know of the following editorial decision as quickly as possible.

As you will see, in its current form, the manuscript is not acceptable for publication in The Journal of Physiology. In comments to me, the Reviewing Editor expressed interest in the potential of this study, but much work still needs to be done (and this may include new experiments) in order to satisfactorily address the concerns raised in the reports.

In view of this interest, I would like to offer you the opportunity to carry out all of the changes requested in full, and to resubmit a new manuscript using the "Submit Special Case Resubmission for JP-RP-2021-281541..." on your homepage.

We cannot, of course, guarantee ultimate acceptance at this stage as the revisions required are substantial. However, we encourage you to consider the requested changes and resubmit your work to us if you are able to complete or address all changes.

A new manuscript would be renumbered and redated, but the original referees would be consulted wherever possible. An additional referee's opinion could be sought, if the Reviewing Editor felt it necessary. A full response to each of the reports should be uploaded with a new version.

I hope that the points raised in the reports will be helpful to you.

Yours sincerely,

Professor Don M. Bers
Senior Editor
The Journal of Physiology
<https://jp.msubmit.net>
<http://jp.physoc.org>
The Physiological Society
Hodgkin Huxley House
30 Farringdon Lane
London, EC1R 3AW
UK
<http://www.physoc.org>
<http://journals.physoc.org>

EDITOR COMMENTS

Reviewing Editor:

Larger sample sizes are required and clarification of the number of samples tested. eg IHC figures

The manuscript has been carefully assessed by 2 reviewers. Both reviewers acknowledge the novel finding and complex experiments but state that further experiments must be undertaken to provide evidence for the role of the Notch -Sma3g cascade. They suggest undertaking studies in spheroids and capillary formation in hypoxic and normoxic conditions. The

findings would be further strengthened by RNA silencing experiments to prove the role of Sema3g in Notch signalling.

REFeree COMMENTS

Referee #1:

In this interesting manuscript the Authors explored signalling pathways which are involved in arterial regeneration following peripheral limb muscle ischemia, in mice. By the use of transgenic adult mice with either gain- or loss-of-function Notch signalling in the vasculature, in vitro cell cultures and transcriptomics analysis the Authors identified Semaphorin3g (Sema3g) as a possible candidate downstream of Notch signalling. Once secreted by arterial endothelial cells, the Authors conclude, Sema3g may assist in functional arterialization and reperfusion of ischemic tissue.

The paper is scientifically sound, the results are significant and this manuscript asks an important physiological question. However, the conclusions should be further consolidated by further experiments.

The findings are interesting and the Authors applied a very sophisticated technical and articulated molecular approach to disentangle different Notch-Sema3g signalling pathways linked to arterial regeneration. The Authors applied a high number of experiments to investigate this, however there are still some points to be resolved and further experiments are required to fully demonstrate their conclusions.

In general, the exposure of the results, although interesting, let me a bit disoriented. During the reading I found some inconsistencies. The role of Notch in angiogenesis and re-vascularisation is already known, and indeed this signalling pathway is very complex since it can either overstimulate or inhibit functional angiogenesis. In this elegant study the Authors identified Semaphorin3g as a master piece for reach a normal vascular pattern, however its role in regeneration after ischemic process should be supported by further in vitro mimicked ischemic experiments.

Indeed, endothelial cells respond differently under normoxia or hypoxia. Under this prospective, Notch -Sma3g cascade in spheroids and capillary formation has to be tested in hypoxic and normoxic conditions. This is a missing information by the Authors and an additional experiment has to be considered.

Finally, to prove the role of Sma3g and properly dissect the Nrp-Sema3g signaling axis in endothelial cells, the Authors have to use RNA silencing. Performing only a qRT-PCR of Nrp and Plexin receptors from confluent GFP expressing HPAEC controls and HPAECs expressing DNMA1L, as they did, is not sufficient to demonstrate the key functional role of Sma3g.

General comments:

1. in all in vivo or ex-vivo experiments the Authors used 5 mice per group. Due to general intraindividual variability, was better have a greater sample size. I understand that this not always can be possible, when working with transgenic mice. Thus, I suggest the authors to express and show all data and statistic not as mean per animal, but raw data per each group.
2. In ischemic and non-ischemic samples is necessary to quantify the level of capillary density or number of capillaries and not just rely on the lumen of the vessels which may simply depend on the more or less internal position of the histological sectioning being examined.
3. Also is not clear if the Authors have any explanation for the increased vessel lumen observed in the arteries after ischemia. It looks more a mechanical adaptation rather than a response linked to increase angiogenesis or capillarisation of collateral vessel. I would expect more an increased in capillary density rather than an increased in a vessel lumen mediated by Notch signalling. How was the capillary density?

Specific comments:

List of abbreviation acronyms: is missing. Please reduced the number of abbreviations, there are too many abbreviations used through the text

Introduction: it needs to be shortened.

-Please amend also the last sentence: where you say: "We further showed Sema3g functioned as a negative regulator of angiogenesis, which s collateral vessels and enables efficient arterial patterning to restore ischemic tissue". ... a part is missing where you say: ", which s collateral vessels and enables"...

Results:

In: Notch signalling during adaptive arteriogenesis and angiogenesis

-I suggest to move in intro the first group of seven lines

Page 5 You say: Immunohistochemistry of collateral vessels in the adductor muscle 7 days post

hindlimb ischemia showed 2-fold increase in the diameter of collateral vessels as compared to the control contralateral limbs without surgery (Fig1D and 1E).

-Ischemic and non-ischemic HIC images are not comparable, the size of nuclei and FCSA are quite different in proportion to came from the same objective magnification. Probably, images were captured using a different objective magnification. Please match the images correctly, with a same magnification. More, a picture showing just two vessels cannot support your conclusion. A large field showing more vessels is mandatory to sustain your data. The graph shows variations in 5 mice per group, but how many vessels have been analysed in each group? how many histological sections were analysed? Please include this information in the figure legends.

In: Inhibition of Notch signaling delays vascular regeneration in vivo

Fig 2 F: main concern. Why do you compare treated transgenic mice versus WT and not DNMA1L transgenic mice versus untreated DNMA1L transgenic mice? Or I did not understand well the aim of this experiment?

In: Notch signaling improves blood flow following hindlimb ischemia

You say vessel lumen of NICD collateral arteries significantly expanded by Day7 following HLI,

in contrast to the arteries of adductors of DNMA1L mice (Fig3D and 3E).

-But again, how can be explained an expansion in the lumen in such a short time, this looks more a mechanical response rather than something related to angiogenesis or neo vascularisation. Did you check whether the vessel cells were proliferating (ki67)? How did you discriminate the neo-forming capillaries from the old resident ones? how many capillaries did you analyse per microscope field? Please give an explanation in the test.

Supplementary Fig1A: the position where side branches were cauterized should be indicated by arrows

Supplementary figure 1C: WB image is not clear, also size of target proteins should be indicated and WB results shown in a graph. Please specify if controls are WT mice or transgenic mice without the treatment. Say just control is a bit ambiguous, in this case.

Figure 3F, if NICD mice are overexpressing Notch, why does the graph show instead any statistical difference between overexpressing or DNMA1L mice?

Suppl.Fig 2A and 2B: please explain and give a better description in figure legends.

Fig 4, panel a to c can be removed or placed as supplementary materials, and please explain the figure contents in figure legends and result text, otherwise these are not understandable by a non-expert reader.

Page 8 You say: Ingenuity pathway analysis suggested the involvement of Sema3g in axonal guidance signaling, where it functioned in concert with other established Notch targets Efn2 and Unc5b (Supplementary Fig3A and 3B).

- I'm sorry but I'm a bit disoriented this is another experiment you did with neuronal cells or is something already knew and

reported from literature? Please clear in the text.

Page 9 end of the paragraph You say: Based on these assessments, which supported our RNA-seq analysis and qRT-PCR validation experiments, we conclude that Sema3g is a downstream target of Notch signaling in endothelial cells.

-You cannot prove this without silencing Sema3g.

In Sema3g functions to inhibit angiogenesis

Page 10 You say: HPAECs expressing DNMA1L produced significantly more sprouts and covered a greater sprouting area than control HPAECs (Fig5B). These findings were consistent with observations where loss of Notch signaling augmented VEGF signaling in endothelial cells, thereby increasing tips cell number (40).

-I'm disoriented again, if this is correct why DNMA1L expressing spheroids responded similarly to SEMA3G treatment and had fewer sprouts and covered less sprouting area than cells treated with vehicle (Fig5A and Fig5B). Sema3g should restore the proper spheroid formation and modulate functional angiogenesis. Sema3g contributes to modulate a functional angiogenesis and normalise vascular pattern.

Page 11 You say: We observed that addition of SEMA3G restored tube formation and increased tube length in DNMA1L expressing HPAECs (SupplementaryFig4C).

-To be honest, some tubes appear to be better with the vehicle, they appear with a homogeneous organisation, in the treated cells they are too uneven in size and shape.

You say: Neuropilin receptors selectively bind to VEGF isoforms as well as Class3 Semaphorins;

thus, to dissect the Nrp-Sema3g signaling axis in endothelial cells, we performed a qRT-PCR of

Nrp and Plexin receptors from confluent GFP expressing HPAEC controls and HPAECs

expressing DNMA1L.

-Again, why did you not use siRNA to dissect the pathways?

You say: Pre-treatment with SEMA3G before the addition of VEGF165 inhibited phosphorylation of VEGFR2 possibly due to the involvement of Nrp2 in Semaphorin signaling (Fig5D).

-this is speculative, until you don't prove this with siRNA

Discussion:

You say: ...GSI washout experiments, where washing away the Notch inhibitor did not result in full restoration of Sema3g expression, which indicate a need of other co-activators to drive for full expression.

-Did you test whether the full restore of Sema3g expression is time dependent, after the Notch inhibitor washout? This could be an explanation. I suggest that a time course response to the washout should be also performed.

You say: "the loss of Notch signaling enhances tip cell generation and should augment revascularization".

-I'm puzzled, again, I see from your results and discussion that the loss of Notch causes hyperformation of tips and therefore an unstable (no functional) capillary network. This means that Notch inhibition does not augment re-vascularisation (instead you say in the sentence that it should) and could explain why in vivo you see a perfusion deficit. Anyway, this conclusion has to be consolidated by an in vitro model of ischemia.

Statistics

Please provide the number of experimental replicates and number of tissue section analysed.

Referee #2:

In this manuscript Hyun and coauthors investigated the potential role of Notch signalling in mediating arterial regeneration. The authors employed various in vivo and cell culture models to verify their hypothesis. In a mouse model of hind limb ischemia the authors demonstrate an upregulation of Jag1/Notch1/Hes1 but not Dll4 and Hey1 on day5. Manoeuvres suppressing the Notch signalling resulted in impaired recovery of vessel perfusion while hyperactivation of Notch signalling did not result in improved/enhanced vessel perfusion compared to WT controls (comparing fig 2C and 3c) although the authors for unknown reasons did not compare NICD (hyperactive Notch) group with WT but with DNMAML (suppressed Notch signalling) group. RNA sequencing data showed differential expression of 3438 genes in endothelial cells derived from both groups. Quite fortunately, the authors were able to identify Sema3G as potential downstream target of Notch signalling.

Using human pulmonary artery endothelial cells (HPAEC) the authors show gamma secretase inhibitor suppresses Notch signalling accompanied by Sema3G suppression. Moreover, Sema3G treatment of HPAECs improves tube formation but suppresses spheroid sprouting in vitro and mouse retian angiogenesis in vivo.

General critic:

The authors start with collateral development which is more arteriogenesis process and shifted to purely angiogenesis. Flow and hypoxia are among the major factors initiating and triggering the development of collaterals, do they have any relation to the mechanism, authors proposed in this study.

Specific:

Dll4 has been shown to play important role in tip/stalk cell phenotype development. This is not changed in the present study but jag1 is upregulated! This should be discussed.

Fig2. Theoretically in DNMAML cells, Notch1 expression should not be reduced, but it is significantly reduced. Would authors like to comment on it ? Does there any feedback mechanism exist?

Are HPAECs comparable to skeletal muscle arterial ECs?

Figure5A/B: Sprout area coverage is not an appropriate way of quantification of sprouting.

The main message (tip/stalk phenotype of ECs) of the study is presented by cell culture experiments which do not depict a true picture of the phenomenon. How the authors can explain the gradients of VEGF/Sema3G and other factors in cell culture? Which type of cells had authors in their cell culture model, tip or stalk?

END OF COMMENTS

Responses to Reviewers' Comments

Referee #1:

In this interesting manuscript the Authors explored signalling pathways which are involved in arterial regeneration following peripheral limb muscle ischemia, in mice. By the use of transgenic adult mice with either gain- or loss-of-function Notch signalling in the vasculature, in vitro cell cultures and transcriptomics analysis the Authors identified Semaphorin3g (Sema3g) as a possible candidate downstream of Notch signalling. Once secreted by arterial endothelial cells, the Authors conclude, Sema3g may assist in functional arterialization and reperfusion of ischemic tissue.

The paper is scientifically sound, the results are significant and this manuscript asks an important physiological question. However, the conclusions should be further consolidated by further experiments.

The findings are interesting and the Authors applied a very sophisticated technical and articulated molecular approach to disentangle different Notch-Sema3g signalling pathways linked to arterial regeneration. The Authors applied a high number of experiments to investigate this, however there are still some points to be resolved and further experiments are required to fully demonstrate their conclusions.

In general, the exposure of the results, although interesting, let me a bit disoriented. During the reading I found some inconsistencies. The role of Notch in angiogenesis and re-vascularisation is already known, and indeed this signalling pathway is very complex since it can either overstimulate or inhibit functional angiogenesis. In this elegant study the Authors identified Semaphorin3g as a master piece for reach a normal vascular pattern, however its role in regeneration after ischemic process should be supported by further in vitro mimicked ischemic experiments.

Indeed, endothelial cells respond differently under normoxia or hypoxia. Under this prospective, Notch -Sma3g cascade in spheroids and capillary formation has to be tested in hypoxic and normoxic conditions. This is a missing information by the Authors and a n additional experiment has to be considered.

Finally, to prove the role of Sma3g and properly dissect the Nrp-Sema3g signaling axis in endothelial cells, the Authors have to use RNA silencing. Performing only a qRT-PCR of Nrp and Plexin receptors from confluent GFP expressing HPAEC controls and HPAECs expressing DNMA1L, as they did, is not sufficient to demonstrate the key functional role of Sma3g.

In response, we appreciated the comments of the reviewer and we apologize for the delay in resubmitting the revised version. The first author had moved to UCLA to take a post doctoral fellowship and hence the delay. We have addressed all the general concerns raised and apologize about the “disoriented” issue raise by the reviewer. We felt the same way as too much information was packed into one manuscript. We hope the revision improved the readability and expect the reviewer will be better able to appreciate the importance of the work. Please see

below for the detailed responses to the comments. All of the revisions have been incorporated into the text and tracked. Overall we hope the manuscript is now far more readable

Reviewers comments are italicized and the responses follow:

General comments:

1. in all in vivo or ex-vivo experiments the Authors used 5 mice per group. Due to general intraindividual variability, was better have a greater sample size. I understand that this not always can be possible, when working with transgenic mice. Thus, I suggest the authors to express and show all data and statistic not as mean per animal, but raw data per each group.

Response: We appreciate the reviewer for understanding the importance of working with transgenic mice. This was essential for drawing more substantive conclusion. In response, we have addressed the concerns above. The IHC measurements of collateral vessels (n=5), are single measurements of a specific collateral vessel that is anatomically located within the neurovascular bundle of the semimembranosus muscle. It is not the average of collateral vessels diameters per animal. The quantification method used was adopted from Limbourg. A “Evaluation of postnatal arteriogenesis and angiogenesis in a mouse model of hind-limb ischemia” Nature Protocols 2009 and in now explained in the Materials and Methods.

2. In ischemic and non-ischemic samples is necessary to quantify the level of capillary density or number of capillaries and not just rely on the lumen of the vessels which may simply depend on the more or less internal position of the histological sectioning being examined.

Response: We unfortunately did not section the tissue in the proper visual plane to enable quantification of capillary densities. Previous studies have established the importance of Notch signaling in angiogenesis which regulates the outgrowth of new capillaries, and specifically the capillaries in the gastrocnemius following hind limb ischemia. Our focus was on Notch signaling and determining Notch targets regulating postnatal vascularization. Because we did not look at capillary density of the adductor muscle, we have accordingly revised this section of the Discussion.

3. Also is not clear if the Authors have any explanation for the increased vessel lumen observed in the arteries after ischemia. It looks more a mechanical adaptation rather than a response linked to increase angiogenesis or capillarisation of collateral vessel. I would expect more an increased in capillary density rather than an increased in a vessel lumen mediated by Notch signalling. How was the capillary density?

Response: Thank you for suggesting mechanical adaptation as a possible mechanism for the expansion of collateral vessels. From what is known, adaptive arteriogenesis is a complex cellular process dependent on cellular and environmental factors. DNMA1L-EC specific mice have been reported to have less basal NO production, making the arteries less susceptible to dilation following occlusion; paper by Chang ACY et al, *Notch Dependent Regulation of the ischemic vasodilatory response* ATVB, 2013. Since collateral arteries are larger than capillaries

and expand and adopt a corkscrew like morphology following vascular occlusion, it is possible that they mount an adaptive response that differs from increased capillary density even though all vessels are Notch deficient.

Specific comments:

List of abbreviation acronyms: is missing. Please reduced the number of abbreviations, there are too many abbreviations used through the text

Response: Abbreviated words have been reduced throughout the text.

Introduction: it needs to be shortened.

Response: The Introduction is shortened

-Please amend also the last sentence: where you say: "We further showed Sema3g functioned as a negative regulator of angiogenesis, which s collateral vessels and enables efficient arterial patterning to restore ischemic tissue". ... a part is missing where you say: ", which s collateral vessels and enables"...

Response: The text has been corrected to reflect our intent.

Results:

In: Notch signalling during adaptive arteriogenesis and angiogenesis

-I suggest to move in intro the first group of seven lines

Page 5 You say: Immunohistochemistry of collateral vessels in the adductor muscle 7 days post hindlimb ischemia showed 2-fold increase in the diameter of collateral vessels as compared to the control contralateral limbs without surgery (Fig1D and 1E).

-Ischemic and non-ischemic HIC images are not comparable, the size of nuclei and FCSA are quite different in proportion to come from the same objective magnification. Probably, images were captured using a different objective magnification. Please match the images correctly, with a same magnification. More, a picture showing just two vessels cannot support your conclusion. A large filed showing more vessels is mandatory to sustain your data. The graph shows variations in 5 mice per group, but how many vessels have been analysed in each group? how many histological sections were analysed? Please include this information in the figure legends.

Response: We appreciate the reviewer for understanding the importance of the work with transgenic mice. The IHC measurements of collateral vessels (n=5), are single measurements of a specific collateral vessel that is anatomically located within the neurovascular bundle of the semimembranosus muscle and not an average of collateral vessels' diameters per animal.

Representative images of matching magnifications have now been included in the figures section.

In: Inhibition of Notch signaling delays vascular regeneration in vivo

Fig 2 F: main concern. Why do you compare treated transgenic mice versus WT and not DNMAHL transgenic mice versus untreated DNMAHL transgenic mice? Or I did not understand well the aim of this experiment?

Response: We regret misleading the reviewer. The aim of this experiment was to understand the increase in Notch signaling in endothelial cells following hindlimb ischemia. This did not occur in DNMAHL mice due to Notch signaling inhibition. WT endothelial cells within adductor muscle responded to ischemia and showed upregulation of components of Notch signaling (Notch1 and Jag1) and the putative downstream target Hes1 compared to endothelial cells in the non-ischemic contralateral adductor. Conversely, Notch deficient endothelial cells (from DNMAHL mice) in the ischemic adductor failed to elicit a Notch response when compared to endothelial cells contained within the non-ischemic contralateral adductor. We did not compare the untreated DNMAHL as those cells would not be exposed to the response following hindlimb ischemia.

In: Notch signaling improves blood flow following hindlimb ischemia

You say vessel lumen of NICD collateral arteries significantly expanded by Day7 following HLI, in contrast to the arteries of adductors of DNMAHL mice (Fig3D and 3E).

-But again, how can be explained an expansion in the lumen in such a short time, this looks more a mechanical response rather than something related to angiogenesis or neo vascularisation. Did you check whether the vessel cells were proliferating (ki67)? How did you discriminate the neo-forming capillaries from the old resident ones? how many capillaries did you analyse per microscope field? Please give an explanation in the test.

Response: It is possible a vasodilatory response resulted from an increase in NO production and improved blood flow. Additionally, 7days following ischemia may be sufficient time for endothelial cells to proliferate and expand from pre-existing collateral arteries. Endothelial cells lining the arterial wall have been shown to have great proliferative potential and can regenerate 0.8mm area within 3 days (McDonald et al). As our research interests were focused on adaptive responses following ischemia and how arterial blood vessels deliver oxygenated blood we narrowed our analysis to arterial microvasculature rather than capillaries. Although we did not analyze capillary density of CD31+ vessels directly, our analysis encompassed CD31+/ α -SMA+ arterial microvasculature (<60um) which feed and branch out into capillaries. This quantification is now included in **Supplementary Figure 1D**. Furthermore, as arterial circulation delivers oxygen-rich blood, we attribute the increase in perfusion to be the result of vessel expansion rather than an increase in capillary density. Remodeling, which is a longer process, ~21days, requires the coordination of cellular processes to deposit ECM and recruit local cell populations such as smooth muscles to accommodate the high flow conditions of arterial circulation (Stabile et al). Establishment of perfusion varies greatly between mouse strains. C57Bl6 mice have been shown to contain pre-existing collaterals and re-establish perfusion quicker than nude and Balb/C and 129S2 mice (Helisch A and Ramo). We have discussed these points in the Discussion.

McDonald A. et al., Endothelial Regeneration of Large Vessels Is a Biphasic Process Driven by Local Cells with Distinct Proliferative Capacities. *Cell Stem Cell* 2018

Ramo K. et al., Suppression of ischemia in arterial occlusive disease by JNK-promoted native collateral artery development. *eLife* 2016;5:e18414

Helisch A. et al., Impact of Mouse Strain Differences in Innate Hindlimb Collateral Vasculature. *ATVB* 2006

Stabile E., et al., Impaired Arteriogenic Response to Acute Hindlimb Ischemia in CD4-Knockout Mice. *Circulation* 2003

Supplementary Fig1A: the position where side branches were cauterized should be indicated by arrows Supplementary Figure1A: clearly denote where vessels were cauterized. Supplementary figure 1C: WB image is not clear, also size of target proteins should be indicated and WB results shown in a graph. Please specify if controls are WT mice or transgenic mice without the treatment. Say just control is a bit ambiguous, in this case.

Response: We thank the reviewer for these suggestions. Cauterized vessels are now shown in solid black lines. Supplementary Figure1B contains a higher resolution image, and the figure legend contains molecular weight of proteins and analysis. Additional details are provided regarding controls have been clarified.

Figure 3F, if NICD mice are overexpressing Notch, why does the graph show instead any statistical difference between overexpressing or DNMAML mice?

Response: NICD mice express the hyperactive intracellular domain of Notch, which enhances Notch signaling by bypassing receptor-ligand interaction and activation. Similar to Figure 2F, Figure 3F demonstrates the fold increases in Notch signaling of endothelial cells in ischemic adductor compared to that of the contralateral limb. Notch signaling is constitutively active in NICD mice therefore the increased signaling was determined by increase of Hey1, a downstream target (Nandagopal et al., 2018, Cell). The differences were compared against DNMAML because the use of NICD mice was a gain of function experiment, thus we wanted to confirm the components and targets of Notch signaling were significantly affected compared against the loss of function model.

Suppl.Fig 2A and 2B: please explain and give a better description in figure legends.

Response: Supplementary Figure 2A and 2B are the gating strategies used to analyze arterial endothelial cells and assess purity of sorted arterial endothelial cells. Pure populations of GFP+ arterial endothelial cells were sorted for bulk RNA sequencing to interrogate the transcriptomic profile of these cells following hindlimb ischemia. This information has been added to the corresponding figure legends.

Fig 4, panel a to c can be removed or placed as supplementary materials, and please explain the figure contents in figure legends and result text, otherwise these are not understandable by a non-expert reader.

Page 8 You say: Ingenuity pathway analysis suggested the involvement of Sema3g in axonal guidance signaling, where it functioned in concert with other established Notch targets Efnb2 and Unc5b (Supplementary Fig3A and 3B).

- I'm sorry but I'm a bit disoriented this is another experiment you did with neuronal cells or is something already known and reported from literature? Please clear in the text.

Response: The above recommended changes are made. Ingenuity pathway analysis is a bioinformatics tool, where differentially expressed genes are categorically sorted based on their assumed functions. Sema3g is a class 3 semaphorin that has been shown to participate in axonal guidance as Semaphorins were originally discovered to provoke neuronal cone collapse. Our confidence in Sema3g as a potential Notch target was further solidified as it was accompanied by Efnb2 and Unc5b, which are well established Notch targets.

Page 9 end of the paragraph You say: Based on these assessments, which supported our RNA-seq analysis and qRT-PCR validation experiments, we conclude that Sema3g is a downstream target of Notch signaling in endothelial cells. You cannot prove this without silencing Sema3g.

Response: Sema3g knockout studies have been conducted by Chen DY. et al., JCI 2021 and Liu X. et al., Cell Rep. 2016, which demonstrate vascular defect and hypersprouting phenotypes that resemble Notch deficiency, and these are the phenotypes we observed. Since these types of studies have been previously carried out we have referenced these findings and described how they support our conclusions.

Page 10 You say: HPAECs expressing DNMAML produced significantly more sprouts and covered a greater sprouting area than control HPAECs (Fig5B). These findings were consistent with observations where loss of Notch signaling augmented VEGF signaling in endothelial cells, thereby increasing tips cell number (40).

-I'm disoriented again, if this is correct why DNMAML expressing spheroids responded similarly to SEMA3G treatment and had fewer sprouts and covered less sprouting area than cells treated with vehicle (Fig5A and Fig5B). Sema3g should restore the proper spheroid formation and modulate functional angiogenesis. Sema3g contributes to modulate a functional angiogenesis and normalise vascular pattern.

Response: The in-vitro system in which DNMAML is over-expressed preferentially generated tip cells at the expense of stalk cells as the normal tip-stalk relationship was dysregulated by inhibition of Notch signaling. Tip cells are inherently more responsive to VEGF as they migrate towards higher concentrations of VEGF to ultimately perfuse ischemic tissue after development

of a functional vascular network. Mechanistically we hypothesize Sema3g acts as negative feedback regulator of VEGF signaling. Therefore, Sema3g partially rescued the hypersprouting phenotype. Further studies assessing the Sema3g dose and synergism with other Semaphorins represent a massive extension of the present studies and they deserve to be studied independently with more rigorous analysis; hence they are outside the scope of our current very comprehensive study.

Page 11 You say: We observed that addition of SEMA3G restored tube formation and increased tube length in DNMAML expressing HPAECs (SupplementaryFig4C). To be honest, some tubes appear to be better with the vehicle, they appear with a homogeneous organisation, in the treated cells they are too uneven in size and shape.

Response. We appreciate the reviewer's insights and have removed these data as they are not relevant to the model and are not required for the conclusions of the studies.

You say: Neuropilin receptors selectively bind to VEGF isoforms as well as Class3 Semaphorins; thus, to dissect the Nrp-Sema3g signaling axis in endothelial cells, we performed a qRT-PCR of Nrp and Plexin receptors from confluent GFP expressing HPAEC controls and HPAECs expressing DNMAML. -Again, why did you not use siRNA to dissect the pathways?

Response: Notch is evolutionary conserved and is involved in multiple aspects of endothelial identity, homeostatic function and recovery. Transient inhibition by siRNA or shRNA is a possible alternative, but our approach of DNMAML utilizes a potent inhibitor of Notch signaling with a specific inhibition of the transcriptional activation mechanism, thus limiting potential and multiple off-target effects. Like Notch, VEGF plays a critical role in endothelial cells, thus we focused on the possible regulatory role of Sema3g on VEGF signaling by acting through the Nrp2-Plexin signaling pathway. This has been clarified in the text.

You say: Pre-treatment with SEMA3G before the addition of VEGF165 inhibited phosphorylation of VEGFR2 possibly due to the involvement of Nrp2 in Semaphorin signaling (Fig5D). this is speculative, until you don't prove this with siRNA

Response: We concur, and this statement is deleted from the text.

Discussion: You say: ...GSI washout experiments, where washing away the Notch inhibitor did not result in full restoration of Sema3g expression, which indicate a need of other co-activators to drive for full expression. -Did you test whether the full restore of Sema3g expression is time dependent, after the Notch inhibitor washout? This could be an explanation. I suggest that a time course response to the washout should be also performed. You say: "the loss of Notch signaling enhances tip cell generation and should augment revascularization". I'm puzzled, again, I see from your results and discussion that the loss of Notch causes hyperformation of tips and therefore an unstable (no functional) capillary network. This means that Notch inhibition does not augment re-vascularisation (instead you say in the sentence that it should) and could explain why in vivo you see a perfusion deficit. Anyway, this conclusion has to be consolidated by an in

vitro model of ischemia.

Response: We have extensively revised the Discussion and taken into consideration each of the points raised above.

Statistics. Please provide the number of experimental replicates and number of tissue section analysed.

Response: All statistical details are provided in the Legends and specifically highlighted in the revised manuscript.

Referee #2:

In this manuscript Hyun and coauthors investigated the potential role of Notch signalling in mediating arterial regeneration. The authors employed various in vivo and cell culture models to verify their hypothesis. In a mouse model of hind limb ischemia the authors demonstrate an upregulation of Jag1/Notch1/Hes1 but not Dll4 and Hey1 on day5. Manoeuvres suppressing the Notch signalling resulted in impaired recovery of vessel perfusion while hyperactivation of Notch signalling did not result in improved/enhanced vessel perfusion compared to WT controls (comparing fig 2C and 3c) although the authors for unknown reasons did not compare NICD (hyperactive Notch) group with WT but with DNMMML (suppressed Notch signalling) group. RNA sequencing data showed differential expression of 3438 genes in endothelial cells derived from both groups. Quite fortunately, the authors were able to identify Sema3G as potential downstream target of Notch signalling.

Using human pulmonary artery endothelial cells (HPAEC) the authors show gamma secretase inhibitor suppresses Notch signalling accompanied by Sema3G suppression. Moreover, Sema3G treatment of HPAECs improves tube formation but suppresses spheroid sprouting in vitro and mouse retina angiogenesis in vivo.

General critic:

The authors start with collateral development which is more arteriogenesis process and shifted to purely angiogenesis. Flow and hypoxia are among the major factors initiating and triggering the development of collaterals, do they have any relation to the mechanism, authors proposed in this study.

Response: This is a very pertinent point. Our study attempts to identify factors that regulate vascularization via arteriogenesis in vivo following ischemic injury. To this end, we isolated arterial endothelial cells from the adductor muscle after ischemic injury and performed RNA sequencing to identify factors downstream of Notch signaling. We believe that this approach clearly considers factors that are influenced by blood flow and hypoxia during in vivo recovery

to post-ischemic injury. However, for a more mechanistic analysis of the downstream genetic target, we focused on the importance of Sema3G. At this stage our experimental strategy shifted toward well-established assays that better assess the function of Sema3G in promoting the formation of the vascular network and fine-tuning vascularization as described in the text. We would be last to say that all elements of Sema3G are understood by our studies or the other related semaphorins (which were identified as shown in the Results) are not involved. Future work will make use of transgenic mouse models that suppress Sema3G levels as well as levels of other semaphorin identified, thus allowing us to determine the multifarious role of these guidance peptides in ordered vascularization in vivo through modulation of Notch signaling.

Specific:

Dll4 has been shown to play important role in tip/stalk cell phenotype development. This is not changed in the present study but jag1 is upregulated! This should be discussed.

Response: The importance of Jag1 has been included in the Discussion.

Fig2. Theoretically in DNAM1L cells, Notch1 expression should not be reduced, but it is significantly reduced. Would authors like to comment on it ?

Response: Regulation of Notch1 in endothelial cells may be similar to that of other cells where Notch1 is a target of Notch signaling (Weng A. et al., Genes Dev. 2006). Notch has been shown to form super enhancer complexes, which can regulate target genes 12.5kb away from the transcriptional sites (Jia Y., Chng W., Zhou J., J Hematol Oncol. 2019)

Does there any feedback mechanism exist?

Response: This is a good point. Perturbation of Notch signaling, both gain- or loss-of-function, have been associated with defects in angiogenesis. Potential feedback mechanisms have been included in the Discussion and Results.

Are HPAECs comparable to skeletal muscle arterial ECs?

Response: Our previous work and that of others demonstrated endothelial cells in culture behave similarly. We chose HPAECs as they are exposed to the high shear stress. The other point is that cultured endothelial cells, from wherever they are derived, are highly plastic and change their phenotype accordingly. We are of the opinion that the origin of the cells is not important but rather their health, plasticity potential, and confluency of the endothelial monolayers affect their overall function.

Figure5A/B: Sprout area coverage is not an appropriate way of quantification of sprouting.

Response: As sprouting endothelial cells can develop radially and span different planes, we utilized GFP to demark all endothelial cells rather than trace just the spheroid. Our approach is adapted to what was used in Aldabbous L., et al ATVB 2016.

The main message (tip/stalk phenotype of ECs) of the study is presented by cell culture experiments which do not depict a true picture of the phenomenon. How the authors can explain the gradients of VEGF/Sema3G and other factors in cell culture?

Response: We agree that in vitro systems cannot fully depict the complexity of in vivo response, however we utilized a defined a valid reductionistic in vitro system to simulate the angiogenic responses. Use of endothelial cells in this type work is common practice since it enables the reduction of complicating and confounding factors and allowed to address questions in a simpler system. Results from the cultured systems recapitulated the vivo findings. We were also able to use minimum amounts of the pro-angiogenic factors, VEGF and FgF, to elicit cellular responses address questions related to the role of Sem 3g. We made the reasonable assumption that in-vitro an initial gradient of VEGF/Sema3g was established in the spheroids and the surrounding environment. Over time levels of VEGF/Sema3g equilibrated within the spheroid, thus we limited our analysis to 24hrs where VEGF mediated vascular sprouts were observed. Furthermore the results concerning the role of Sema3g in the spheroids were fully confirmed by studying the retinal vessels, which has been classically used to address the orderly formation of a vascular network

Which type of cells had authors in their cell culture model, tip or stalk?

Response: We utilized both tip and stalk cells as both are needed to study the role Notch signaling. Tip cells respond to pro-angiogenic factors, which specify orientation and directionality, while the stalks cells proliferate and extend the outgrowth of vessels. In order to capture both cells types for qRT-PCR analysis we utilized endothelial cells almost at fully confluency.

Dear Dr Malik,

Re: JP-RP-2021-282286X "Notch1 Promotes Revascularization through Modulation of Downstream Vascular Patterning Signaling" by James Hyun, Monica Y Lee, Jalees Rehman, Kostandin V Pajcini, and Asrar B Malik

Thank you for submitting your manuscript to The Journal of Physiology. It has been assessed by a Reviewing Editor and by 2 expert Referees and I am pleased to tell you that it is considered to be acceptable for publication following satisfactory revision.

The reports are copied at the end of this email. Please address all of the points and incorporate all requested revisions, or explain in your Response to Referees why a change has not been made.

NEW POLICY: In order to improve the transparency of its peer review process The Journal of Physiology publishes online as supporting information the peer review history of all articles accepted for publication. Readers will have access to decision letters, including all Editors' comments and referee reports, for each version of the manuscript and any author responses to peer review comments. Referees can decide whether or not they wish to be named on the peer review history document.

Authors are asked to use The Journal's premium BioRender (<https://biorender.com/>) account to create/redrawn their Abstract Figures. Information on how to access The Journal's premium BioRender account is here: <https://physoc.onlinelibrary.wiley.com/journal/14697793/biorender-access> and authors are expected to use this service. This will enable Authors to download high-resolution versions of their figures.

I hope you will find the comments helpful and have no difficulty returning your revisions within 4 weeks.

Your revised manuscript should be submitted online using the links in Author Tasks Link Not Available.

Any image files uploaded with the previous version are retained on the system. Please ensure you replace or remove all files that have been revised.

REVISION CHECKLIST:

- Article file, including any tables and figure legends, must be in an editable format (eg Word)
- Abstract figure file (see above)
- Statistical Summary Document
- Upload each figure as a separate high quality file
- Upload a full Response to Referees, including a response to any Senior and Reviewing Editor Comments;
- Upload a copy of the manuscript with the changes highlighted.

- A potential 'Cover Art' file for consideration as the Issue's cover image;
- Appropriate Supporting Information (Video, audio or data set https://jp.msubmit.net/cgi-bin/main.plex?form_type=display_requirements#supp).

To create your 'Response to Referees' copy all the reports, including any comments from the Senior and Reviewing Editors, into a Word, or similar, file and respond to each point in colour or CAPITALS and upload this when you submit your revision.

I look forward to receiving your revised submission.

If you have any queries please reply to this email and staff will be happy to assist.

Yours sincerely,

Professor Don M. Bers

Senior Editor
The Journal of Physiology
<https://jp.msubmit.net>
<http://jp.physoc.org>
The Physiological Society
Hodgkin Huxley House
30 Farringdon Lane
London, EC1R 3AW
UK
<http://www.physoc.org>
<http://journals.physoc.org>

REQUIRED ITEMS:

-You must start the Methods section with a paragraph headed Ethical Approval. A detailed explanation of journal policy and regulations on animal experimentation is given in Principles and standards for reporting animal experiments in The Journal of Physiology and Experimental Physiology by David Grundy J Physiol, 593: 2547-2549. doi:10.1113/JP270818.). A checklist outlining these requirements and detailing the information that must be provided in the paper can be found at: <https://physoc.onlinelibrary.wiley.com/hub/animal-experiments>. Authors should confirm in their Methods section that their experiments were carried out according to the guidelines laid down by their institution's animal welfare committee, and conform to the principles and regulations as described in the Editorial by Grundy (2015). The Methods section must contain details of the anaesthetic regime: anaesthetic used, dose and route of administration and method of killing the experimental animals.

-The Reference List must be in Journal format

-Your manuscript must include a complete Additional Information section

-The Journal of Physiology funds authors of provisionally accepted papers to use the premium BioRender site to create high resolution schematic figures. Follow this link and enter your details and the manuscript number to create and download figures. Upload these as the figure files for your revised submission. If you choose not to take up this offer we require figures to be of similar quality and resolution. If you are opting out of this service to authors, state this in the Comments section on the Detailed Information page of the submission form.

-Please upload separate high-quality figure files via the submission form.

-You must upload original, uncropped western blot/gel images (including controls) if they are not included in the manuscript. This is to confirm that no inappropriate, unethical or misleading image manipulation has occurred <https://physoc.onlinelibrary.wiley.com/hub/journal-policies#imagmanip> These should be uploaded as 'Supporting information for review process only'. Please label/highlight the original gels so that we can clearly see which sections/lanes have been used in the manuscript figures.

-Your paper contains Supporting Information of a type that we no longer publish. Any information essential to an understanding of the paper must be included as part of the main manuscript and figures. The only Supporting Information that we publish are video and audio, 3D structures, program codes and large data files. Your revised paper will be returned to you if it does not adhere to our Supporting Information Guidelines

-Papers must comply with the Statistics Policy https://jp.msubmit.net/cgi-bin/main.plex?form_type=display_requirements#statistics

In summary:

-If $n < 30$, all data points must be plotted in the figure in a way that reveals their range and distribution. A bar graph with data points overlaid, a box and whisker plot or a violin plot (preferably with data points included) are acceptable formats.

-If $n > 30$, then the entire raw dataset must be made available either as supporting information, or hosted on a not-for-profit repository e.g. FigShare, with access details provided in the manuscript.

- n clearly defined (e.g. x cells from y slices in z animals) in the Methods. Authors should be mindful of pseudoreplication.

-All relevant n values must be clearly stated in the main text, figures and tables, and the Statistical Summary Document (required upon revision)

-The most appropriate summary statistic (e.g. mean or median and standard deviation) must be used. Standard Error of the Mean (SEM) alone is not permitted.

-Exact p values must be stated. Authors must not use 'greater than' or 'less than'. Exact p values must be stated to three significant figures even when 'no statistical significance' is claimed.

-Statistics Summary Document completed appropriately upon revision

-A Data Availability Statement is required for all papers reporting original data. This must be in the Additional Information section of the manuscript itself. It must have the paragraph heading "Data Availability Statement". All data supporting the results in the paper must be either: in the paper itself; uploaded as Supporting Information for Online Publication; or archived in an appropriate public repository. The statement needs to describe the availability or the absence of shared data. Authors must include in their Statement: a link to the repository they have used, or a statement that it is available as Supporting Information; reference the data in the appropriate section(s) of their manuscript; and cite the data they have shared in the References section. Whenever possible the scripts and other artefacts used to generate the analyses presented in the paper should also be publicly archived. If sharing data compromises ethical standards or legal requirements then authors are not expected to share it, but must note this in their Statement. For more information, see our Statistics Policy.

-Please include an Abstract Figure. The Abstract Figure is a piece of artwork designed to give readers an immediate understanding of the research and should summarise the main conclusions. If possible, the image should be easily 'readable' from left to right or top to bottom. It should show the physiological relevance of the manuscript so readers can assess the importance and content of its findings. Abstract Figures should not merely recapitulate other figures in the manuscript. Please try to keep the diagram as simple as possible and without superfluous information that may distract from the main conclusion(s). Abstract Figures must be provided by authors no later than the revised manuscript stage and should be uploaded as a separate file during online submission labelled as File Type 'Abstract Figure'. Please ensure that you include the figure legend in the main article file. All Abstract Figures should be created using BioRender. Authors should use The Journal's premium BioRender account to export high-resolution images. Details on how to use and access the premium account are included as part of this email.

EDITOR COMMENTS

Reviewing Editor:

The revised manuscript has been carefully reviewed and there remain some concerns regarding the images provided and detail regarding analysis of the data. The authors are asked to respond to both reviewers' concerns and provide additional/alternative images and clarification regarding analysis where requested.

REFEREE COMMENTS

Referee #1:

By adding the changes, the Authors highly improved the quality of the manuscript and its readability. However, some points required a better clarification

1. my previous comment

Results Page 5: You say: Immunohistochemistry of collateral vessels in the adductor muscle 7 days post hindlimb ischemia showed 2-fold increase in the diameter of collateral vessels as compared to the control contralateral limbs without surgery (Fig1D and 1E).

"Ischemic and non-ischemic HIC images are not comparable, the size of nuclei and FCSA are quite different in proportion to came from the same objective magnification. Probably, images were captured using a different objective magnification. Please match the images correctly, with a same magnification. More, a picture showing just two vessels cannot support your conclusion. A large field showing more vessels is mandatory to sustain your data. The graph shows variations in 5 mice per group, but how many vessels have been analyzed in each group? how many histological sections were analyzed? Please include this information in the figure legends.

Your reply:

"Response: We appreciate the reviewer for understanding the importance of the work with transgenic mice. The IHC measurements of collateral vessels (n=5), are single measurements of a specific collateral vessel that is anatomically located within the neurovascular bundle of the semimembranosus muscle and not an average of collateral vessels' diameters per animal" Representative images of matching magnifications have now been included in the figures section.

I appreciate your clarification, but HIC (H&E) images of matching magnifications look removed and not included in Fig. 1D. There is only a representative epifluorescence immunostaining in Fig. 1D, and not the original HIC (removed). Epifluorescence images, although fine, show only a zoomed-in enlargement of a selected region, where are visible only two vessel lumens, and not a larger panoramic of the cross section; is not possible therefore appreciate the capillary bed morphology and distinguish the neurovascular bundle. I recommend to include a representative HIC staining acquired with a lower magnification in order to make visible the capillary bed and the neurovascular bundle, and indeed mark the neurovascular bundle and the position of the vessel you analyzed.

2. Results Figure 3F

my previous comment: "if NICD mice are over-expressing Notch, why does the graph show instead any statistical difference between over-expressing or DNMAML mice?"

Your reply:

Response: NICD mice express the hyperactive intracellular domain of Notch, which enhances Notch signaling by bypassing receptor-ligand interaction and activation. Similar to Figure 2F, Figure 3F demonstrates the fold increases in Notch signaling of endothelial cells in ischemic adductor compared to that of the controlateral limb. Notch signaling is constitutively active in NICD mice therefore the increased signaling was determined by increase of Hey1, a downstream target (Nandagopal et al., 2018, Cell). The differences were compared against DNMAML because the use of NICD mice was a gain of function experiment, thus we wanted to confirm the components and targets of Notch signaling were significantly affected compared against the loss of function model.

Sorry again, but my concern is why did you normalize the data to contralateral non-ischemic wild-type controls. Since the aim was to compare ischemic DNMAML vs ischemic NICD (overexpressing Notch), I'm wondering why did you normalize the data vs non-ischemic WT, this approach could add bias to your results. If you wanted to normalise the data against a non-ischemic control, I would have rather used the muscle of the contralateral leg from the same animal. So, I strongly invite to revise this graph, and show the not normalized expression data obtained by qRT-PCR to exclude any possible bias that could detract quality to the results or their interpretation. I would suggest to show a graph (or in a table..) with expression analysis of WT, DNMAML, and NICD ischemic and non-ischemic groups.

Referee #2:

Before discussing the authors' rebuttal, this reviewer would like to highlight a discrepancy pertaining to the animal data presentation.

The DNMAML day 7 image in figure 2B is same as DNMAML day 7 image in Fig 3B.

The ischemic PCR data (blue columns) in Figure 1F are exactly same as in figure 2F (WT)

The PCR data in figure 2F DNMAML (red columns) is exactly same as in Figure 3F (DNMAML, red columns).

The DNMAML column (red) in figure 2G is same as in Figure 3G (DNMAML, red).

This suggests the authors have performed only a single set of experiments and data are re-used from one figure in the other figure. This, however, is not clearly stated in the manuscript and wrongly shows as the authors have performed multiple sets of experiments.

Otherwise, the authors addressed the critique adequately.

END OF COMMENTS

1st Confidential Review

27-Aug-2021

EDITOR COMMENTS

Reviewing Editor:

The revised manuscript has been carefully reviewed and there remain some concerns regarding the images provided and detail regarding analysis of the data. The authors are asked to respond to both reviewers' concerns and provide additional/alternative images and clarification regarding analysis where requested.

REFEREE COMMENTS

Referee #1:

By adding the changes, the Authors highly improved the quality of the manuscript and its readability. However, some points required a better clarification

1. my previous comment

Results Page 5: You say: Immunohistochemistry of collateral vessels in the adductor muscle 7 days post hindlimb ischemia showed 2-fold increase in the diameter of collateral vessels as compared to the control contralateral limbs without surgery (Fig1D and 1E).

"Ischemic and non-ischemic HIC images are not comparable, the size of nuclei and FCSA are quite different in proportion to come from the same objective magnification. Probably, images were captured using a different objective magnification. Please match the images correctly, with a same magnification. More, a picture showing just two vessels cannot support your conclusion. A large field showing more vessels is mandatory to sustain your data. The graph shows variations in 5 mice per group, but how many vessels have been analyzed in each group? how many histological sections were analyzed? Please include this information in the figure legends.

Your reply:

"Response: We appreciate the reviewer for understanding the importance of the work with transgenic mice. The IHC measurements of collateral vessels (n=5), are single measurements of a specific collateral vessel that is anatomically located within the neurovascular bundle of the semimembranosus muscle and not an average of collateral vessels' diameters per animal" Representative images of matching magnifications have now been included in the figures

section.

I appreciate your clarification, but HIC (H&E) images of matching magnifications look removed and not included in Fig. 1D. There is only a representative epifluorescence immunostaining in Fig. 1D, and not the original HIC (removed). Epifluorescence images, although fine, show only a zoomed-in enlargement of a selected region, where are visible only two vessel lumens, and not a larger panoramic of the cross section; is not possible therefore appreciate the capillary bed morphology and distinguish the neurovascular bundle. I recommend to include a representative HIC staining acquired with a lower magnification in order to make visible the capillary bed and the neurovascular bundle, and indeed mark the neurovascular bundle and the position of the vessel you analyzed.

We thank the reviewer for the suggestions. We have included a representative lower magnification image stained by H&E. The neurovascular bundle has been annotated to show the nerve, vein, and artery. A zoomed in higher magnification of the neurovascular bundle has also been included and a description of the magnification has been included in the figure legends.

2. Results Figure 3F

my previous comment: "if NICD mice are over-expressing Notch, why does the graph show instead any statistical difference between over-expressing or DNMAML mice?"

Your reply:

Response: NICD mice express the hyperactive intracellular domain of Notch, which enhances Notch signaling by bypassing receptor-ligand interaction and activation. Similar to Figure 2F, Figure 3F demonstrates the fold increases in Notch signaling of endothelial cells in ischemic adductor compared to that of the controlateral limb. Notch signaling is constitutively active in NICD mice therefore the increased signaling was determined by increase of Hey1, a downstream target (Nandagopal et al., 2018, Cell). The differences were compared against DNMAML because the use of NICD mice was a gain of function experiment, thus we wanted to confirm the components and targets of Notch signaling were significantly affected compared against the loss of function model.

Sorry again, but my concern is why did you normalize the data to contralateral non-ischemic wild-type controls. Since the aim was to compare ischemic DNMAML vs ischemic NICD (overexpressing Notch), I'm wondering why did you normalize the data vs non-ischemic WT, this approach could add bias to your results. If you wanted to normalise the data against a non-ischemic control, I would have rather used the muscle of the contralateral leg from the same animal. So, I strongly invite to revise this graph, and show the not normalized expression data

obtained by qRT-PCR to exclude any possible bias that could detract quality to the results or their interpretation. I would suggest to show a graph (or in a table..) with expression analysis of WT, DNMAML, and NICD ischemic and non-ischemic groups.

We thank the reviewer for critically reviewing the data. All the data points presented for the qRT-PCR analysis were normalized to the contralateral non-ischemic limb of each animal. The experimental aim was to validate and correlate changes in target gene expression with our Notch inhibition and overexpression model. We have incorporated the reviewer's suggestion and revised the graph to now show the fold change of expression of arterial endothelial cells in WT, DNMAML, and NICD.

Referee #2:

Before discussing the authors' rebuttal, this reviewer would like to highlight a discrepancy pertaining to the animal data presentation.

The DNMAML day 7 image in figure 2B is same as DNMAML day 7 image in Fig 3B.

The ischemic PCR data (blue columns) in Figure 1F are exactly same as in figure 2F (WT)

The PCR data in figure 2F DNMAML (red columns) is exactly same as in Figure 3F (DNMAML, red columns).

The DNMAML column (red) in figure 2G is same as in Figure 3G (DNMAML, red).

This suggests the authors have performed only a single set of experiments and data are re-used from one figure in the other figure. This, however, is not clearly stated in the manuscript and wrongly shows as the authors have performed multiple sets of experiments.

We thank the reviewer for the comments regarding repeat data presentation.

We have now explicitly stated in the figure legends which sets of experiments were re-used for comparison and statistical analysis. The day7 DNMAML images in Fig3A and Fig4B are the same representative image of 8 different experimental animals. We believe the same image assists the reader and better highlights the differences in perfusion between the genotypes. Collateral artery measurements, PCR data, and aEC recovery analysis have now been consolidated and presented in Fig4 to again highlight the differences between each genotype.

Otherwise, the authors addressed the critique adequately.

END OF COMMENTS

Dear Dr Malik,

Re: JP-RP-2021-282286XR1 "Notch1 Promotes Revascularization through Modulation of Downstream Vascular Patterning Signaling" by James Hyun, Monica Y Lee, Jalees Rehman, Kostandin V Pajcini, and Asrar B Malik

I am pleased to tell you that your paper has been accepted for publication in The Journal of Physiology, subject to any modifications to the text and/or satisfactory clarification of the Methods section that may be required by the Journal Office to conform to House rules.

NEW POLICY: In order to improve the transparency of its peer review process The Journal of Physiology publishes online as supporting information the peer review history of all articles accepted for publication. Readers will have access to decision letters, including all Editors' comments and referee reports, for each version of the manuscript and any author responses to peer review comments. Referees can decide whether or not they wish to be named on the peer review history document.

The last Word version of the paper submitted will be used by the Production Editors to prepare your proof. When this is ready you will receive an email containing a link to Wiley's Online Proofing System. The proof should be checked and corrected as quickly as possible.

Authors should note that it is too late at this point to offer corrections prior to proofing. Major corrections at proof stage, such as changes to figures, will be referred to the Reviewing Editor for approval before they can be incorporated. Only minor changes, such as to style and consistency, should be made a proof stage. Changes that need to be made after proof stage will usually require a formal correction notice.

All queries at proof stage should be sent to TJP@wiley.com

The accepted version of the manuscript will be published online, prior to copy editing, in the Accepted Articles section.

Are you on Twitter? Once your paper is online, why not share your achievement with your followers. Please tag The Journal (@jphysiol) in any tweets and we will share your accepted paper with our 22,000+ followers!

Yours sincerely,

Professor Don M. Bers
Senior Editor
The Journal of Physiology
<https://jp.msubmit.net>
<http://jp.physoc.org>
The Physiological Society
Hodgkin Huxley House
30 Farringdon Lane
London, EC1R 3AW
UK
<http://www.physoc.org>
<http://journals.physoc.org>

P.S. - You can help your research get the attention it deserves! Check out Wiley's free Promotion Guide for best-practice recommendations for promoting your work at www.wileyauthors.com/eeo/guide. And learn more about Wiley Editing Services which offers professional video, design, and writing services to create shareable video abstracts, infographics, conference posters, lay summaries, and research news stories for your research at www.wileyauthors.com/eeo/promotion.

* IMPORTANT NOTICE ABOUT OPEN ACCESS *

Information about Open Access policies can be found here <https://physoc.onlinelibrary.wiley.com/hub/access-policies>

To assist authors whose funding agencies mandate public access to published research findings sooner than 12 months after publication The Journal of Physiology allows authors to pay an open access (OA) fee to have their papers made freely available immediately on publication.

You will receive an email from Wiley with details on how to register or log-in to Wiley Authors Services where you will be able to place an OnlineOpen order.

You can check if your funder or institution has a Wiley Open Access Account here <https://authorservices.wiley.com/author-resources/Journal-Authors/licensing-and-open-access/open-access/author-compliance-tool.html>

Your article will be made Open Access upon publication, or as soon as payment is received.

If you wish to put your paper on an OA website such as PMC or UKPMC or your institutional repository within 12 months of publication you must pay the open access fee, which covers the cost of publication.

OnlineOpen articles are deposited in PubMed Central (PMC) and PMC mirror sites. Authors of OnlineOpen articles are permitted to post the final, published PDF of their article on a website, institutional repository, or other free public server, immediately on publication.

Note to NIH-funded authors: The Journal of Physiology is published on PMC 12 months after publication, NIH-funded authors DO NOT NEED to pay to publish and DO NOT NEED to post their accepted papers on PMC.

EDITOR COMMENTS

Reviewing Editor:

There are minor corrections that need attention. The manuscript is acceptable for publication.

REFEREE COMMENTS

Referee #1:

Please amend the following typos:

Figure 1: image C. Please check the correct title position of the low magnification images on the left side of the panel. Image on the left: top title, non-ischemic; bottom title: ischemic

pg.8 (revised text, in red): there is a duplicate .. and and...

pg 35: G, missing bracket...(n=5 animals.

Referee #2:

The authors responded my comments adequately.

END OF COMMENTS

2nd Confidential Review

28-Oct-2021